# State Space Prompting via Gathering and Spreading Spatio-Temporal Information for Video Understanding

**Jiahuan Zhou[1], Kai Zhu[1], Zhenyu Cui[1], Zichen Liu[1], Xu Zou[2]\*, Gang Hua[3]**

[1]Wangxuan Institute of Computer Technology, Peking University, Beijing 100871, China
[2]Huazhong University of Science and Technology, Wuhan 430074,China
[3]Amazon.com, Inc, Bellevue, WA 98004, USA

jiahuanzhou@pku.edu.cn, zhukai2022@ruc.edu.cn
{cuizhenyu,lzc20180720}@stu.pku.edu.cn, zx@zoux.me, ganghua@gmail.com

## Abstract

Recently, pre-trained state space models have shown great potential for video classification, which sequentially compresses visual tokens in videos with linear complexity, thereby improving the processing efficiency of video data while maintaining high performance. To apply powerful pre-trained models to downstream tasks, prompt learning is proposed to achieve efficient downstream task adaptation with only a small number of fine-tuned parameters. However, the sequentially compressed visual prompt tokens fail to capture the spatial and temporal contextual information in the video, thus limiting the effective propagation of spatial information within a video frame and temporal information between frames in the state compression model and the extraction of discriminative information. To tackle the above issue, we proposed a State Space Prompting (SSP) method for video understanding, which combines intra-frame and inter-frame prompts to aggregate and propagate key spatiotemporal information in the video. Specifically, an Intra-Frame Gathering (IFG) module is designed to aggregate spatial key information within each frame. Besides, an Inter-Frame Spreading (IFS) module is designed to spread discriminative spatio-temporal information across different frames. By adaptively balancing and compressing key spatio-temporal information within and between frames, our SSP effectively propagates discriminative information in videos in a complementary manner. Extensive experiments on four video benchmark datasets verify that our SSP significantly outperforms existing SOTA methods by 2.76% on average while reducing the overhead of fine-tuning parameters. The code is available at https://github.com/zhoujiahuan1991/NeurIPS2025-SSP.

## 1 Introduction

In recent years, the Vision Transformer (ViT) has demonstrated its promising performance in video understanding due to its powerful attention-based context modelling capabilities [1, 2, 3, 4, 5, 6, 7, 8, 9, 10]. However, the computational cost of the attention mechanism, which increases quadratically with the length of the input data, incurs huge computational and memory costs, especially when processing long video sequences. To achieve efficient video processing, a state space modelling method, called VideoMamba, is introduced to achieve comparable performance to the ViT while maintaining linear computational complexity [11, 12]. Despite some progress in model pre-training[13, 14, 15], it still suffers from the heavy overhead of downstream task adaptation through parameter fine-tuning. Therefore, Parameter-Efficient Fine-Tuning (PEFT) has aroused extensive attention to achieve

---

\*Corresponding author

39th Conference on Neural Information Processing Systems (NeurIPS 2025).

comparable or even higher performance than Full Fine-Tuning (FFT) and reduce the adaptation costs by optimizing only a few set of parameters [16, 17, 18].

Early PEFT methods mainly focused on parameter-level efficient fine-tuning, *e.g.,* Adapter [19, 20, 21, 22] or LoRA [23, 24], but still suffered from inefficiency with the additional introduction of model parameters. To this end, visual prompt learning technology aims to embed a small set of learnable prompting tokens at the input level, enabling efficient downstream task adaptation without extending internal model parameters [25, 26, 27, 28, 29, 30, 31, 32, 33, 34]. However, as shown in Figure 1(a), existing video prompting methods are typically ViT-oriented, which exploit global attention mechanisms to propagate key information in intra- and inter-frame prompts [35, 36, 37, 38], ignoring the requirement to balance efficiency and effectiveness in video understanding for VideoMamba. Specifically, ignoring the sequential compression state space [39, 40, 41, 42, 43], existing video prompting methods fail to gather spatial information in long video token sequences for Mamba. In addition, the global attention-oriented prompting method mitigates the high efficiency in VideoMamba, leading to the dissemination of key spatial and temporal information in the long video token sequence. Consequently, as shown in Figure 1(b), the embedded prompts mitigate the aggregation of discriminative spatio-temporal information due to the information decay after long-term compression [44].

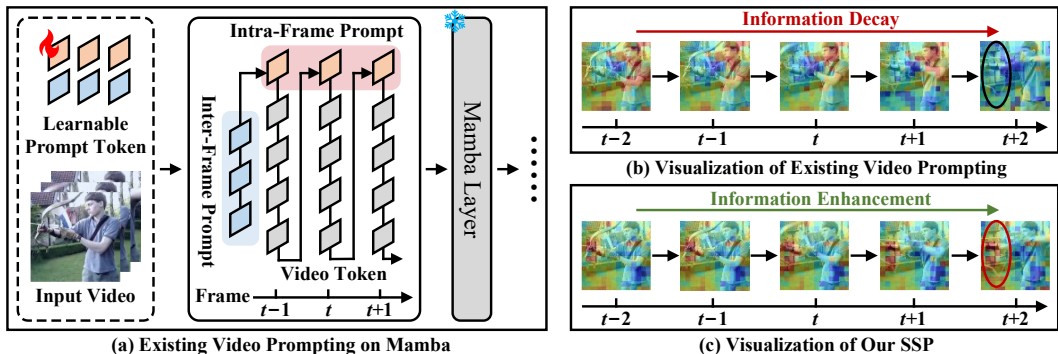

Figure 1: Existing video prompting methods on Mamba and its visualization results of updating gate compared to our SSP. Existing methods directly concatenate learnable prompts to video tokens, resulting in the information decay problem after the long-term state space compression. However, our SSP achieves information enhancement through spatial gathering and temporal spreading.

To address the above challenges, we propose a State Space Prompting (SSP) framework for video understanding, which gathers and spreads spatio-temporal information in an efficient and effective manner, as shown in Figure 1(c). Specifically, an Intra-Frame Gathering (IFG) module is designed to exploit a low-rank local convolution to aggregate spatial information within each video frame. Sequentially, a low-rank attention-oriented Inter-Frame Spreading (IFS) module is further proposed to spread key information at the temporal level, which develops a low-rank attention module to refine the temporal information that is gradually aggregated between frames, where long-term context information is effectively spread within global temporal prompts. Among them, the information entropy of each frame adjusts the attention given to each frame when refining the temporal information, and the frame-specific spatial variance is employed to gate the influence strength from the spreading information to each frame.

In summary, our contributions are three-fold: (1) We proposed a State Space Prompting method for Video Understanding, called SSP, which gathers and spreads discriminative spatio-temporal information compressed by the state space model to achieve high effectiveness while maintaining its computational efficiency. (2) We design an Intra-Frame Gathering module and an Inter-Frame Spreading module to facilitate spatio-temporal contextual information interaction by spreading gathered local spatial information in a temporal manner. (3) Extensive experiments on multiple video understanding benchmarks demonstrate that our method achieves superior performance against existing methods with only ∼3% of tunable parameters compared to full tuning.

## 2 Related Work

### 2.1 State Space Model (SSM)

In recent years, State Space Models (SSMs) have emerged as a promising approach for sequence modeling, offering the ability to capture long-range dependencies with linear computational complexity [39, 40, 41, 42, 43]. Building on this foundation, Mamba and Mamba2 introduced input-dependent update and forget gates to address limitations in content-based reasoning [45, 46]. Unlike Transformer architectures, this allows SSM parameters to be dynamically modulated by input, significantly improving expressiveness in discrete modalities. Coupled with hardware-friendly parallelization, these advances lead to notable gains in computational efficiency and performance in language modeling.

Building on this progress, Vision Mamba extended the Mamba framework to 2D image modeling via bidirectional spatial scanning, achieving strong results in image understanding tasks [47, 48]. More recently, VideoMamba further generalized this approach to video by introducing spatio-temporal scanning, enabling efficient modeling of global dependencies across both spatial and temporal dimensions with linear complexity [11, 49]. This architecture rivals or surpasses traditional CNN and Transformer models [12, 50, 51, 52], establishing itself as a competitive backbone for video understanding. However, applying pre-trained models like VideoMamba to downstream tasks using full fine-tuning typically requires large data and high computational cost. Thus, developing parameter-efficient fine-tuning strategies for VideoMamba remains an important and urgent challenge.

### 2.2 Parameter-Efficient Fine-Tuning for SSM

Parameter-efficient fine-tuning techniques aim to reduce learnable parameters while maintaining model performance, thereby reducing storage and computational costs when adapting pre-trained models to downstream tasks [16]. Several studies have attempted to apply parameter-efficient fine-tuning methods to Mamba architecture models [53, 54, 55]. These methods can be categorized into partial-based, addition-based, and prompt-based approaches.

**Partial-based methods** typically fine-tune only a subset of parameters in the pre-trained Mamba model, such as projection layers, convolution layers, or forget gates [54, 55]. These methods are straightforward and easy to implement. However, partial-based approaches are constrained by the model's inherent parameter space, limiting their adaptability to downstream tasks across different domains. **Addition-based methods** generally freeze the original model parameters and incorporate learnable components, such as adapter modules [19, 21]. While these plug-and-play modules can be readily transferred to the Mamba model architecture, they simply apply transformations to the input data without considering the sequential progression characteristics inherent to Mamba's architecture. The Additional-Scan approach attempts to learn downstream knowledge by increasing the state dimensions of the SSM [55]. However, when applied to video domains, merely increasing state dimensions proves insufficient for effectively extracting critical spatio-temporal information. **Prompt-based methods** add a small number of learnable prompt vectors, optimizing only these parameters during training. Existing work such as SVP has transferred prompt learning methods to Mamba models by generating prompts for each token, effectively activating Mamba's update and forget gates during fine-tuning [53]. However, such methods are designed exclusively for static 2D images. When transferred to video tasks, they similarly struggle with the challenge of modeling long-context spatio-temporal relationships.

### 2.3 Video Prompting

Prompt learning methods were first introduced in natural language processing (NLP) to transfer pre-trained models to various downstream tasks [56, 57, 58, 59]. Inspired by the success of prompt learning in NLP, these methods have been extended to the visual domain[60, 61]. VPT and VFPT fine-tune models by concatenating token-based prompts with input data to capture image features [25, 26]. More recently, methods like DGL and STOP have applied prompt learning to video tasks, using intra-frame and inter-frame prompt modules to capture temporal information [35, 36, 37, 38]. For instance, DGL employs prompt vectors as query, key and value vectors to model both local features and global features in videos [36]. MPT utilizes prompts as query vectors, leveraging the Q-Former mechanism to extract spatial, temporal, and global features [37]. STOP generates inter-frame prompts

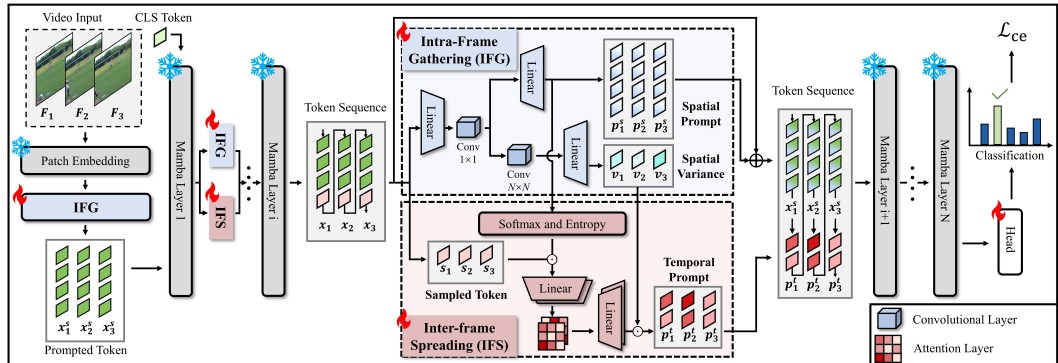

Figure 2: The pipeline of our SSP. We embed videos into image tokens with spatial prompts. After the initial Mamba layer, our complementary IFG and IFS modules operate - IFG aggregates spatial information while IFS spreads temporal information. Information entropy and frame-specific spatial variance bridge these modules. The prompted tokens and CLS token then pass through subsequent Mamba layers for video classification.

from all tokens while using intra-frame prompts to highlight the varying importance of frames, dynamically inserting prompts between frames [38].

However, these methods were originally designed for the Transformer architecture, leveraging its unique structure by connecting prompts to query and key-value vectors to store task information. Transformer-based video prompting methods use a global attention mechanism [62], allowing tokens within the sequence to interact at any position. In contrast, the Mamba architecture propagates tokens sequentially [63, 64, 65], causing adjacent tokens to contain more overlapping information. As a result, when applying existing video prompting methods to Mamba, feeding all tokens into the inter-frame prompt module introduces redundant contextual information. This issue is more pronounced in video data with high spatiotemporal redundancy, making it difficult for the prompt module to effectively capture and propagate discriminative spatiotemporal context, ultimately limiting the model's performance.

# 3 Methodology

In this section, we illustrate the proposed SSP comprehensively, and the overall pipeline is depicted in Figure 2.

## 3.1 Preliminary of Mamba

The SSM-based models, Mamba, Vision Mamba (ViM), and VideoMamba, are inspired by continuous systems that map one-dimensional equations or sequences $x(t) \in \mathbb{R} \mapsto y(t) \in \mathbb{R}$ through a $D$-dimensional hidden state $h(t) \in \mathbb{R}^D$. These hidden states evolve over time via parameter matrices $\mathbf{A}$, $\mathbf{B}$, and $\mathbf{C}$, following a linear ordinary differential equation:

$$h'(t) = \mathbf{A}h(t) + \mathbf{B}x(t), \quad y(t) = \mathbf{C}h(t), \tag{1}$$

where parameter $\mathbf{A} \in \mathbb{R}^{D \times D}$ represents the forgetting gate matrix, $\mathbf{B} \in \mathbb{R}^{D \times 1}$ denotes the update gate matrix, and $\mathbf{C} \in \mathbb{R}^{1 \times D}$ serves as the output projection matrix.

To facilitate application in deep learning, SSMs are discretized into discrete-time systems using the zero-order hold technique. The continuous parameters $\mathbf{A}$ and $\mathbf{B}$ are transformed into their discrete counterparts $\overline{\mathbf{A}} \in \mathbb{R}^{D \times D}$ and $\overline{\mathbf{B}} \in \mathbb{R}^{D \times 1}$, employing a sampling time interval $\Delta \in \mathbb{R}$:

$$\overline{\mathbf{A}} = \exp(\Delta\mathbf{A}), \quad \overline{\mathbf{B}} = (\Delta\mathbf{A})^{-1}(\exp(\Delta\mathbf{A}) - I) \cdot \Delta\mathbf{B}. \tag{2}$$

Consequently, the discretized SSM can be expressed as follows:

$$h_i = \overline{\mathbf{A}}h_{i-1} + \overline{\mathbf{B}}x_i, \quad y_i = \mathbf{C}h_i, \tag{3}$$

where $h_{i-1}, h_i \in \mathbb{R}^{D \times d}$ and $x_i, y_i \in \mathbb{R}^{1 \times d}$, $d$ is the dimension of the input sequences.

## 3.2 State Space Prompting

The backbone of our method is VideoMamba [49], the input is a video $\boldsymbol{V} \in \mathbb{R}^{T \times C \times H \times W}$, where $T$ represents the number of frames, $C$ represents the number of channels, and $H \times W$ is spatial size. Each video frame $\{\boldsymbol{F}_i\}_{i=1}^T$ is split into $N = \frac{H \times W}{h \times w}$ fixed-size patches of size $h \times w$ and these patches are flattened into a sequence of vectors $\boldsymbol{I}_i = \{\boldsymbol{I}_{ij} \in \mathbb{R}^{C \times h \times w}\}_{j=1}^N$, where $i$ denotes the frame index while $j$ denotes the patch index. These vectors are then projected into input tokens $\boldsymbol{x}_i = \{\boldsymbol{x}_{ij}\}_{j=1}^N$, where $\boldsymbol{x}_{ij} \in \mathbb{R}^d$, and $d$ is the hidden dimension of the input sequence. For video classification tasks, the class token $\boldsymbol{x}_{cls}$ is prepended to the sequence of input tokens, which is $[\boldsymbol{x}_{cls}, \boldsymbol{x}_1, \boldsymbol{x}_2, \dots, \boldsymbol{x}_T]$. Then the input tokens are fed into the VideoMamba backbone, which consists of $L$ layers of Mamba block. The class token $\boldsymbol{x}_{cls}$ from the last layer is used for classification tasks. The final output of the VideoMamba backbone is obtained by applying a linear classifier head on $\boldsymbol{x}_{cls}$.

Our SSP consists of two complementary modules: an intra-frame gathering module (IFG) and an inter-frame spreading module (IFS). These modules interact complementarily to aggregate spatial information and spread discriminative long-term context information at the temporal level during fine-tuning.

### 3.2.1 Intra-Frame Gathering Module

The IFG $\mathcal{P}^s$ processes each frame $\boldsymbol{F}_i$ to generate intra-frame prompts $\boldsymbol{p}_i^s \in \mathbb{R}^{N \times d}$, information entropy weights $\boldsymbol{w}_i \in \mathbb{R}^{1 \times d}$, and spatial variance measurements $\boldsymbol{v}_i \in \mathbb{R}^{1 \times d}$. The intra-frame prompts are then overlaid to the input tokens to produce spatial prompted tokens $\boldsymbol{x}_i^s \in \mathbb{R}^{N \times d}$:

$$\boldsymbol{p}_i^s, \boldsymbol{w}_i, \boldsymbol{v}_i = \mathcal{P}^s(\boldsymbol{x}_i), \quad \boldsymbol{x}_i^s = \boldsymbol{x}_i + \boldsymbol{p}_i^s. \tag{4}$$

The IFG takes the tokens from each frame $\boldsymbol{x}_i = \{\boldsymbol{x}_{ij}\}_{j=1}^N$ as input. These tokens are processed through a downsampling layer $\mathcal{L}_1^{down}$ followed by a 2D convolutional layer $\mathcal{N}_1^s$ to generate low-rank feature maps $\boldsymbol{l}_i \in \mathbb{R}^{N \times d^s}$ for each frame, where $d^s$ represents the internal dimension of the intra-frame gathering module. The low-rank feature maps $\boldsymbol{l}_i$ are subsequently upsampled via a linear layer $\mathcal{L}_1^{up}$ to match the dimensionality of the input tokens, resulting in intra-frame prompts $\boldsymbol{p}_i^s$. Concurrently, the low-rank feature maps undergo additional 2D convolution and upsampling operations to produce spatial variance $\boldsymbol{v}_i$. The intra-frame prompts $\boldsymbol{p}_i^s$ are then fed into an entropy calculation module $\mathcal{E}$ to calculate information entropy weights $\boldsymbol{w}_i$:

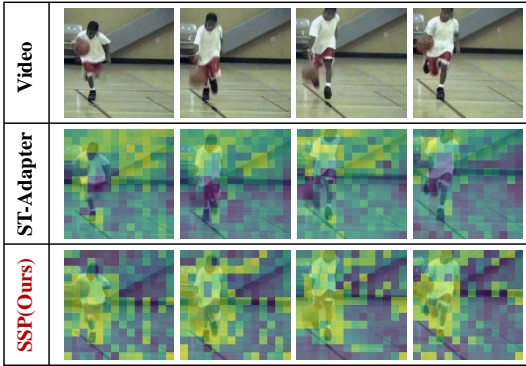

Figure 3: Visualization of our intra-frame prompts, which capture discriminative local features.

$$\begin{aligned} \boldsymbol{l}_i &= \mathcal{N}_1^s(\mathcal{L}_1^{down}(\boldsymbol{x}_i)), & \boldsymbol{p}_i^s &= \mathcal{L}_1^{up}(\boldsymbol{l}_i), \\ \boldsymbol{v}_i &= \mathcal{L}_2^{up}(\mathcal{N}_2^s(\boldsymbol{l}_i)), & \boldsymbol{w}_i &= \mathcal{E}(\boldsymbol{p}_i^s). \end{aligned} \tag{5}$$

The entropy calculation module $\mathcal{E}$ evaluates the informational significance of intra-frame prompts and generates frame-level weights. This module first transforms the intra-frame prompts $\boldsymbol{p}_i^s$ into probability distributions, then calculates the information entropy, which is subsequently scaled by a learnable factor $\alpha$. A small positive constant $\epsilon$ is incorporated to prevent taking the logarithm of zero:

$$P_i = \mathrm{softmax}(\boldsymbol{p}_i^s), H_i = -\sum_d P_i \cdot \log(P_i + \epsilon), E_i = 1.0 - \frac{H_i}{\max(H_i)}, \tag{6}$$

$$\boldsymbol{w}_i = \alpha \cdot \mathrm{softmax}(\bar{E}), \quad \text{where } \bar{E} \text{ is the mean of } E_i \text{ across tokens per frame.}$$

The intra-frame prompts $\boldsymbol{p}_i^s$ gather the model's attention to local features during downstream fine-tuning, as shown in Figure 3. Information entropy weights $\boldsymbol{w}_i$ adjust the attention given to each frame during inter-frame prompt generation based on the certainty of information distribution, while spatial variance $\boldsymbol{v}_i$ gates the influence strength from the long-term context information to local features.

### 3.2.2 Inter-Frame Spreading Module

After processing through the first Mamba layer, the inter-frame spreading module (IFS) activates. This module first samples the last token $s_i \in \mathbb{R}^{1 \times d}$ from frame $F_i$ in the Mamba forward scanning sequence. This token is Hadamard multiplied with the information entropy weights $w_i$ generated by the intra-frame gathering module and fed into the inter-frame spreading module $\mathcal{P}^t$. The module's output is then Hadamard multiplied with the spatial variance $v_i$ to produce inter-frame prompts $p_i^t \in \mathbb{R}^{1 \times d}$:

$$s_i = x_{iN}, \quad p_i^t = \mathcal{P}^t(s_i \odot w_i) \odot v_i. \tag{7}$$

The IFS processes the input through a sequence of operations including a downsampling linear transformation $\mathcal{L}_2^{down}$, an attention computation $\mathcal{A}$, and an upsampling linear transformation $\mathcal{L}_3^{up}$. The resulting output is then computed Hadamard product with the spatial variance $v_i$, and subsequently scaled by a learnable factor $\beta$ to produce the inter-frame prompts $p_i^t$. This architecture facilitates the spreading of long-term context information across video frames while preserving local context information:

$$p_i^t = \beta \cdot \mathcal{L}_3^{up}(\mathcal{A}(\mathcal{L}_2^{down}(s_i))) \odot v_i. \tag{8}$$

The last token of each frame $s_i$ aggregates contextual information from both the current frame and preceding frames during forward scanning, as well as subsequent frames during backward scanning. As shown in Figure 4, the generated inter-frame prompts $p_i^t$ represent temporal inductive biases that spread the gathered temporal information from all frames to the current frame.

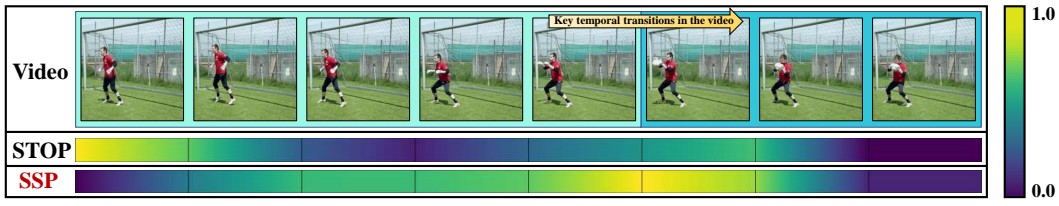

Figure 4: Visualization of our inter-frame prompts. The action "catch" can be parsed into two distinct phases: preparation for the catch and the actual reception of the ball. Our inter-frame prompts effectively locate the key transitional moments between these critical phases.

Subsequently, the inter-frame prompts $p_i^t$ are concatenated with the spatial prompted tokens $x_i^s$, along with the class token $x_{cls}$, are fed into the following $L - 1$ Mamba layers $\{\mathcal{B}_j\}_{j=2}^L$ to extract spatio-temporal features. Inspired by VPT-deep [25], we overlay the intra-frame prompt on the input of each layer and embed the inter-frame prompt in the input of all layers after the first layer. To obtain the final predicted probability distribution $y$, we apply a linear classification head $\mathcal{H}$ on $x_{cls}$.

### 3.2.3 Overall Optimization

As mentioned above, our SSP introduces only a few additional parameters:

$$\mathcal{M} = \{\mathcal{P}^s, \mathcal{P}^t\}. \tag{9}$$

Following prior works, we keep the pre-trained model frozen during training, allowing only the classification head $\mathcal{H}$ and the newly added modules $\mathcal{M}$ to be trainable. The optimization objective is defined as follows:

$$\underset{\mathcal{M}, \mathcal{H}}{\arg\min} \; \mathcal{L}_{ce}(y, y_{gt}), \tag{10}$$

where $\mathcal{L}_{ce}$ is the cross-entropy loss, and $y_{gt}$ is the ground truth video label.

## 4 Experiments

### 4.1 Datasets

**HMDB51** [67] contains 6849 clips across 51 action categories, with an average duration of 3.15 seconds and 91.49 frames per video. It was collected from various sources, mostly from movies, and a small proportion from public databases such as the Prelinger archive, YouTube and Google videos, featuring diverse real-world actions with variations in background and camera angles.

Table 1: The comparison results on K400 pretrained VideoMamba-S (Parameters 25.42M).

| | Method | Venue | Param | HMDB51 | SSV2 | UCF101 | Breakfast |
|---|---|---|---|---|---|---|---|
| VideoMamba-S | Full [49] | *ECCV'24* | 25.42M | 67.58 | 58.57 | 92.96 | 94.27 |
| | Adapter [19] | *NeurIPS'22* | 2.40M | 73.79 | 36.45 | 94.18 | 84.89 |
| | ST-Adapter [21] | *NeurIPS'22* | 2.69M | 70.52 | 30.94 | 94.87 | 77.60 |
| | VPT [25] | *ECCV'22* | 1.50M | 72.74 | 30.68 | 95.16 | 81.25 |
| | VFPT [26] | *NeurIPS'24* | 1.50M | 72.41 | 30.37 | 95.08 | 79.68 |
| | SVP [53] | *AAAI'25* | 2.76M | 69.93 | 38.01 | 95.58 | 80.72 |
| | Additional-Scan [55] | *ICLR'25* | 0.66M | 73.20 | 33.71 | 95.63 | 78.64 |
| | STOP [38] | *CVPR'25* | 1.49M | 70.06 | 21.22 | 93.44 | 65.62 |
| | SSP(Ours) | *This Paper* | 0.98M | **74.38** | **38.68** | **95.69** | **85.41** |

Table 2: The comparison results on CLIP-400M pretrained CLIP-ViT-B/32 [66] (Parameters 88.00M) and K400 pretrained VideoMamba-M (Parameters 74.00M).

| | Method | Venue | Param | HMDB51 | SSV2 | UCF101 | Breakfast |
|---|---|---|---|---|---|---|---|
| CLIP | DGL-Linear [36] | *AAAI'24* | 0.83M | 67.20 | 18.30 | 92.50 | - |
| | DGL-Transformer [36] | *AAAI'24* | 9.57M | 69.80 | 18.10 | 93.60 | - |
| | STOP [38] | *CVPR'25* | 7.53M | 72.00 | 21.40 | 95.30 | - |
| VideoMamba-M | Full [49] | *ECCV'24* | 74.00M | 76.30 | 67.30 | 96.00 | 95.31 |
| | Adapter [19] | *NeurIPS'22* | 2.40M | 73.59 | 46.98 | 96.24 | 89.06 |
| | ST-Adapter [21] | *NeurIPS'22* | 2.69M | 71.96 | 31.91 | 94.81 | 67.70 |
| | VPT [25] | *ECCV'22* | 1.50M | 73.59 | 40.58 | 95.45 | 81.77 |
| | VFPT [26] | *NeurIPS'24* | 1.50M | 71.89 | 39.68 | 95.71 | 86.97 |
| | SVP [53] | *AAAI'25* | 2.76M | 73.66 | 49.08 | 96.77 | 90.10 |
| | Additional-Scan [55] | *ICLR'25* | 1.33M | 73.52 | 44.65 | 96.32 | 86.97 |
| | STOP [38] | *CVPR'25* | 1.49M | 71.96 | 23.44 | 94.60 | 71.35 |
| | SSP(Ours) | *This Paper* | 2.41M | **76.66** | **53.72** | **97.03** | **93.23** |

**UCF101** [68] is an action recognition data set of realistic action videos, collected from YouTube, having 13320 video clips across 101 action categories. The average video length is 7.21 seconds with 186.5 frames per video.

**Something-Something V2** (SSV2) [69] is a large-scale dataset for action recognition, containing 220,847 videos across 174 action categories. The dataset is designed to capture fine-grained actions and interactions between objects, with an average video length of 3.82 seconds and 45.84 frames.

**Breakfast** [70] is a long video understanding dataset containing 1989 video clips divided into 10 categories related to breakfast preparation. The dataset has an average video length of 137.53 seconds, with an average of 2062.89 frames per video.

## 4.2 Comparison Methods

We compare our SSP with both adapter-based and prompt-based parameter-efficient finetuning methods. We also report the fully tuning results as a baseline, i.e., VideoMamba [49]. For adapter-based methods, we compare with the following methods: Adapter [19] and ST-Adapter [21]. For prompt-based methods, we compare with the following methods: DGL-Linear [36], DGL-Transformer [36], VPT [25], VFPT [26], SVP [53], and STOP [38].

## 4.3 Implementation Details

Following [49], all video frames are resized to $224 \times 224$ and split into $14 \times 14$ patches. For HMDB51, UCF101 and SSV2 datasets, each video is uniformly sampled to 8 frames, while for Breakfast, we sample 32 frames. We set the learning rates to 3e-3, 5e-3, 2e-4, and 1e-3 for HMDB51, UCF101,

SSV2, and Breakfast respectively. Meanwhile, we set the batch size to 32 for HMDB51 and UCF101, 64 for Breakfast and 512 for SSV2. All the dataset splits are consistent with the official annotation files. The model is fine-tuned with the AdamW optimizer on 4 NVIDIA 4090-24G GPUs, with a cosine decay scheduler. Additionally, we adopt a warm-up strategy within the first 5 training epochs.

## 4.4 Comparison with State-of-the-arts

We evaluated our SSP method on four popular video datasets: HMDB51, UCF101, SSV2, and Breakfast. For fair comparison, we used CLIP-400M pretrained CLIP-ViT-B/32 as the backbone for ViT-oriented methods, while for methods fine-tuned on the Mamba architecture, we employed Kinetics-400 [71] pretrained VideoMamba-M and VideoMamba-S as backbones. Our method achieves state-of-the-art performance across all datasets, as shown in Table 2.

When fine-tuned on VideoMamba-M, our approach attains top-1 accuracies of 76.66%, 53.72%, 97.03%, and 93.23% on HMDB51, Something-Something V2, UCF101, and Breakfast respectively, representing improvements of **3.00%**, **4.64%**, **0.26%**, and **3.13%** over existing parameter-efficient video fine-tuning methods. As demonstrated in Table 1, even with the parameter constraints of smaller-scale models, our method still delivers superior results when fine-tuned on VideoMamba-S. This effectiveness stems from our tailored design for the VideoMamba architecture, which enables efficient gathering and spreading of discriminative spatio-temporal information within state space models, through the complementary IFG and IFS modules. Notably, our approach demonstrates particularly significant improvements on the challenging large-scale Something-Something V2 dataset and the long-video Breakfast dataset, as shown in Figure 5,

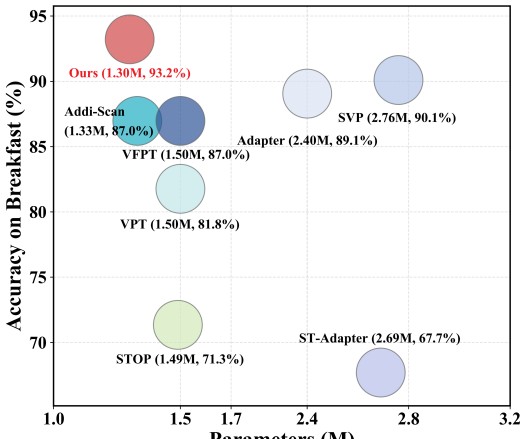

Figure 5: Comparison on Breakfast. SSP outperforms existing methods while tuning minimal parameters.

our method exceeds others when tuning only 1.30M parameters (using only one IFS module as shown in Figure 7). This is because our SSP can gather local information on key regions in complex video data and spread critical global information in long contexts.

## 4.5 Ablation

### 4.5.1 Influence of Different Components

To verify the effectiveness of the intra-frame gathering (IFG) module and inter-frame spreading (IFS) module, we conducted ablation experiments on three datasets: HMDB51, UCF101, and Breakfast, as shown in Table 3. As demonstrated, when neither component is used, SSP achieves the lowest accuracy on all datasets. When the intra-frame gathering module is used alone, the model's performance improves by 9.78% on average. This is because the intra-frame spatial prompts effectively capture local features and enhance the model's ability to focus on discriminative information within each frame. When the inter-frame spreading module is used alone, the model's performance improves by 9.20% on average. This can be attributed to that the inter-frame temporal prompts effectively aggregate and spread global contextual information across frames. When both components are used together, the model achieves the best performance by an average improvement of 13.46% across all datasets, as they complement each other in gathering and spreading both local and global information. When the spatial variance gate or the entropy gate (denoted as $v_i$, $w_i$ in Equation 7) is removed, the model's performance drops by 2.33% and 2.27% respectively on average across all datasets. This indicates that both gates play a crucial role in facilitating the inter-frame prompts to propagate spatial information in a complementary manner based on the key local features of each frame.

Table 3: Ablation of different prompting modules and gates.

| Different Prompting Modules | | | | | Different Gates | | | | |
|---|---|---|---|---|---|---|---|---|---|
| IFG | IFS | HMDB51 | UCF101 | Breakfast | Entropy | Spatial | HMDB51 | UCF101 | Breakfast |
| - | - | 59.34 | 90.64 | 76.56 | - | - | 74.70 | 96.56 | 89.58 |
| ✔ | - | 74.44 | 96.56 | 84.89 | ✔ | - | 75.09 | 96.29 | 88.54 |
| - | ✔ | 72.61 | 96.14 | 85.41 | - | ✔ | 74.96 | 96.61 | 88.54 |
| ✔ | ✔ | **76.66** | **97.03** | **93.23** | ✔ | ✔ | **76.66** | **97.03** | **93.23** |

### 4.5.2 The Visualization Results of Update Gate

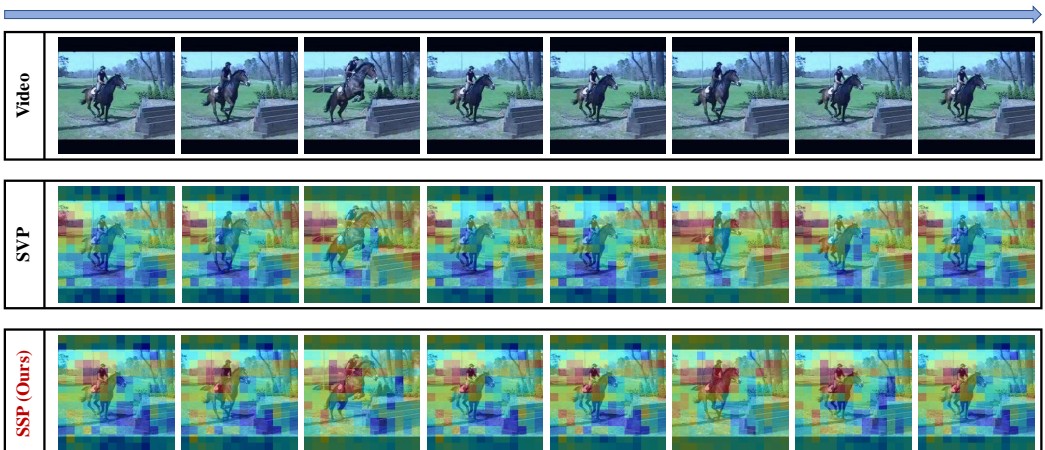

Figure 6: Visualization of the update gate. In SSP, the key region is activated effectively.

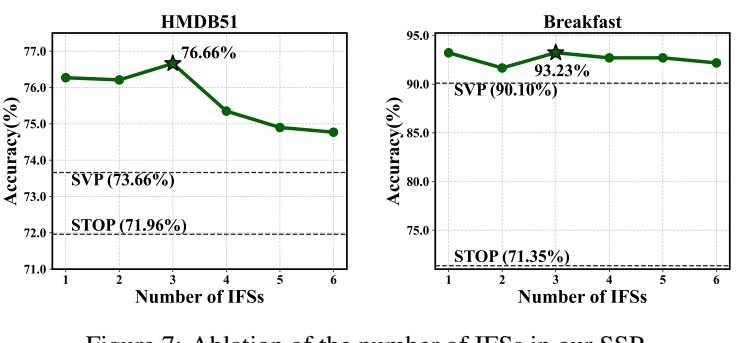

Figure 7: Ablation of the number of IFSs in our SSP.

To further explore the impact of our intra-frame gathering module and inter-frame spreading module, we visualize the normalized update gate values in VideoMamba on the last layer, which contribute to the final classification results directly. As shown in Figure 6, existing prompting methods designed for Mamba like SVP can't effectively gather and spread the discriminative spatio-temporal information. This caused the pre-trained VideoMamba model to focus on irrelevant regions in each frame. As a result, the model struggles to accurately understand dynamic key features in the video. In contrast, our SSP method highlight the key regions with dynamic changes in the video, leading to a more accurate understanding of the video content.

### 4.5.3 Influence of Hyperparameters

The number of the inter-frame spreading modules ($\mathcal{P}^t$) is one of important hyperparameters in our method. To assess its impact, we conduct extensive ablation experiments. As shown in Figure 7, the model's performance initially improves but then fluctuates as the number of inter-frame spreading modules increases.

To balance performance and tunable parameters, we set the number of inter-frame spreading modules to 3 when using VideoMamba-M as backbone, and when using VideoMamba-S as backbone, we set the number to 1.

## 5 Conclusion

In this paper, we propose a novel State Space Prompting (SSP) approach for efficient adaptation of pre-trained state space models to video understanding tasks. SSP combines complementary intra-frame gathering module and inter-frame spreading module to aggregate key spatial information within frames and spread discriminative temporal information between frames, enabling effective propagation of crucial spatio-temporal features in state space models. By employing the two modules, we can adaptively balance and compress discriminative information in videos. The effectiveness of our proposed SSP has been validated on four video benchmarks compared to other parameter-efficient fine-tuning methods.

## 6 Acknowledgements

This work was supported by the National Natural Science Foundation of China (62376011) and the National Key R&D Program of China (2024YFA1410000).

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

# A   Discussion and Analysis

In this section, we discuss and analyze the effectiveness of our SSP method for parameter-efficient fine-tuning the VideoMamba architecture in downstream task adaptation.

For each Mamba layer's input, we denote the $i$-th input token along the scanning order as $\boldsymbol{x}_i$. The hidden state computation for the subsequent Mamba layer is formulated as:

$$\boldsymbol{h}_i = \overline{\mathbf{A}}_i \boldsymbol{h}_{i-1} + \overline{\mathbf{B}}_i \boldsymbol{x}_i^s, \quad \boldsymbol{x}_i^s = \boldsymbol{x}_i + \boldsymbol{p}_i^s, \tag{11}$$

where $\boldsymbol{h}_i$ is the $i$-th compressed hidden state token of the Mamba layer. In the Mamba architecture, each layer's parameter matrices $\mathbf{B} \in \mathbb{R}^{D \times 1}$, $\mathbf{C} \in \mathbb{R}^{1 \times D}$, and $\Delta \in \mathbb{R}$ are derived from the input token $\boldsymbol{x}_i^s$ through functions $\mathcal{S}_B$, $\mathcal{S}_C$, and $\mathcal{S}_\Delta$, respectively:

$$\overline{\mathbf{A}}_i = \exp(\Delta \mathbf{A}_i), \quad \mathbf{B}_i = \mathcal{S}_B(\boldsymbol{x}_i^s), \quad \mathbf{C}_i = \mathcal{S}_C(\boldsymbol{x}_i^s), \quad \Delta_i = \mathcal{S}_\Delta(\boldsymbol{x}_i^s). \tag{12}$$

In our proposed SSP method, the intra-frame prompt $\boldsymbol{p}_i^s$ enables direct fine-tuning of the parameter matrices generated by each token:

$$\overline{\mathbf{A}}_i^p = \exp(\mathcal{S}_\Delta(\boldsymbol{x}_i + \boldsymbol{p}_i^s)\widetilde{\odot}\mathbf{A}), \quad \overline{\mathbf{B}}_i^p = \mathcal{S}_\Delta(\boldsymbol{x}_i + \boldsymbol{p}_i^s)\mathcal{S}_B(\boldsymbol{x}_i + \boldsymbol{p}_i^s), \tag{13}$$

where $\overline{\mathbf{A}}_i^p \in \mathbb{R}^{D \times D}$ and $\overline{\mathbf{B}}_i^p \in \mathbb{R}^{D \times 1}$ represent the forget and update gates directly controlled by the intra-frame prompt $\boldsymbol{p}_i^s$, facilitating the extraction of locally discriminative information, thereby gathering the spatial information effectively when fine-tuning on downstream tasks.

For the inter-frame prompt $\boldsymbol{p}_j^t$ of the $j$-th frame, this prompt vector aggregates global spatio-temporal information and is gated through Hadamard multiplication with the spatial variance $\boldsymbol{v}_j$. It is inserted between the $j$-th and $(j+1)$-th frames, directly influencing the hidden state at that position:

$$\boldsymbol{h}_{j+1,0} = \overline{\mathbf{A}}_{j+1,0}^p \boldsymbol{h}_{jN} + \overline{\mathbf{B}}_{j+1,0}^p \boldsymbol{p}_j^t, \tag{14}$$

where $N$ denotes the number of tokens per frame. Tokens following the $j$-th frame can access global discriminative information through the hidden state $\boldsymbol{h}_{j+1,0}$ of inter-frame prompt $\boldsymbol{p}_j^t$, overcoming the sequential spatio-temporal information transfer limitation of the original VideoMamba model and achieving efficient gathering and spreading of spatio-temporal information.

Next, we further analyze the impact of inter-frame prompts on long-range spatio-temporal information transmission. By expanding Equation 11, we obtain:

$$\begin{aligned}
\boldsymbol{h}_j &= \overline{\mathbf{A}}_j^p \boldsymbol{h}_{j-1} + \overline{\mathbf{B}}_j^p \boldsymbol{x}_j^s = \overline{\mathbf{A}}_j^p \left( \overline{\mathbf{A}}_{j-1}^p \boldsymbol{h}_{j-2} + \overline{\mathbf{B}}_{j-1}^p \boldsymbol{x}_{j-1}^s \right) + \overline{\mathbf{B}}_j^p \boldsymbol{x}_j^s \\
&= \prod_{k=i+1}^{j} \overline{\mathbf{A}}_k^p \boldsymbol{h}_i + \sum_{t=i+1}^{j} \left( \prod_{k=t+1}^{j} \overline{\mathbf{A}}_k^p \right) \overline{\mathbf{B}}_t^p \boldsymbol{x}_t^s,
\end{aligned} \tag{15}$$

where $i < j$. The coefficient of the first term in the above equation, $\prod_{k=i+1}^{j} \overline{\mathbf{A}}_k^p$, represents the influence strength from the $i$-th hidden state token to the $j$-th hidden state token along the scanning sequence. We denote this as the transmission matrix $\mathbf{T}_{i \to j}$:

$$\mathbf{T}_{i \to j} = \prod_{k=i+1}^{j} \overline{\mathbf{A}}_k^p = \exp\left( \sum_{k=i+1}^{j} \Delta_k \widetilde{\odot} \mathbf{A} \right), \tag{16}$$

where $\mathbf{A}$ is a negative matrix. Examining the equation above, we observe that the mutual influence strength between tokens at different positions in the sequence exponentially decays to zero as the distance $j - i$ increases. Without the insertion of inter-frame prompts, the maximum information transmission path length in the sequence equals the sequence length $\mathcal{O}(TN)$, where $T$ is the number of video frames and $N$ is the number of tokens per frame. However, after inserting inter-frame prompts, information from tokens at different positions can be transmitted through the inter-frame prompts inserted after each frame, reducing the maximum information transmission path length to $\mathcal{O}(N)$. This significant reduction in information transmission path length facilitates the aggregation and propagation of global spatio-temporal information.

# B Visualization of More Cases

## B.1 Update Gate Value Visualization

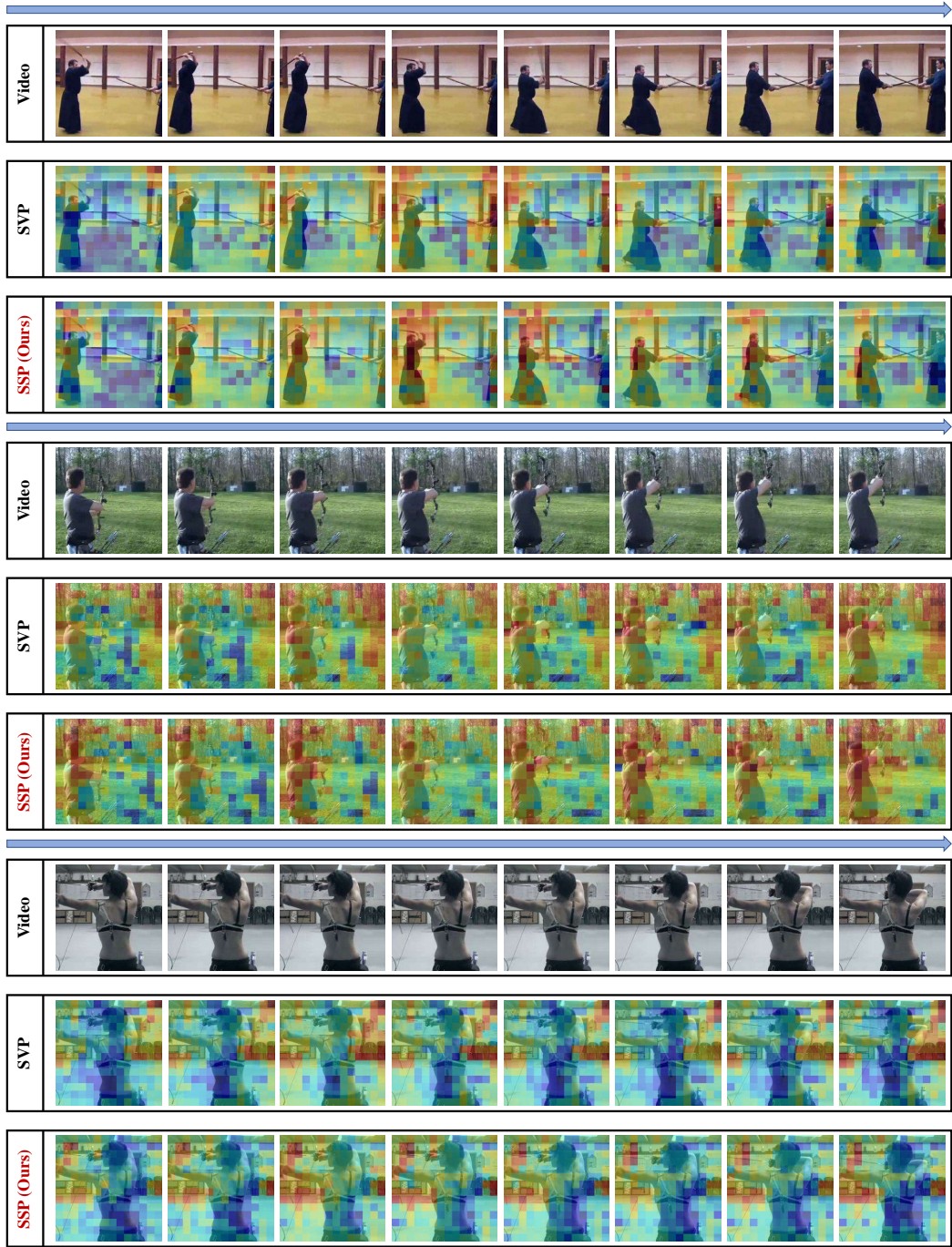

Figure 8: Visualization of the values of the update gate over the last layer in VideoMamba.

The update gate value of each token, denoted as $\overline{\mathbf{B}}_i^p$ in Equation 13, represents the degree of influence each token exerts on the hidden states of VideoMamba. By visualizing the update gate values across tokens, we can observe which specific tokens receive greater attention from the VideoMamba architecture during processing.

As illustrated in Figure 8, existing prompting approaches for Mamba architecture (e.g., SVP) employ static prompts for each frame. This limitation prevents the pre-trained VideoMamba model from effectively integrating and modeling global video contextual information, thereby hindering accurate interpretation of human actions within the video sequence. In contrast, our SSP method, through the complementary application of intra-frame gathering and inter-frame spreading modules, dynamically emphasizes regions exhibiting significant temporal variations in the video. This approach enables more precise comprehension of video content by capturing the most relevant spatio-temporal information.

## B.2 Intra-Frame Prompts Visualization

In our SSP method, the intra-frame prompts, which are overlaid on the input tokens fed into each Mamba layer, are employed to capture the discriminative local features and gather the spatial information of each frame. To visualize the intra-frame prompts, we plot the values of intra-frame prompts of all frames in a video over the last Mamba layer as heatmaps, and overlay them on the original video frames.

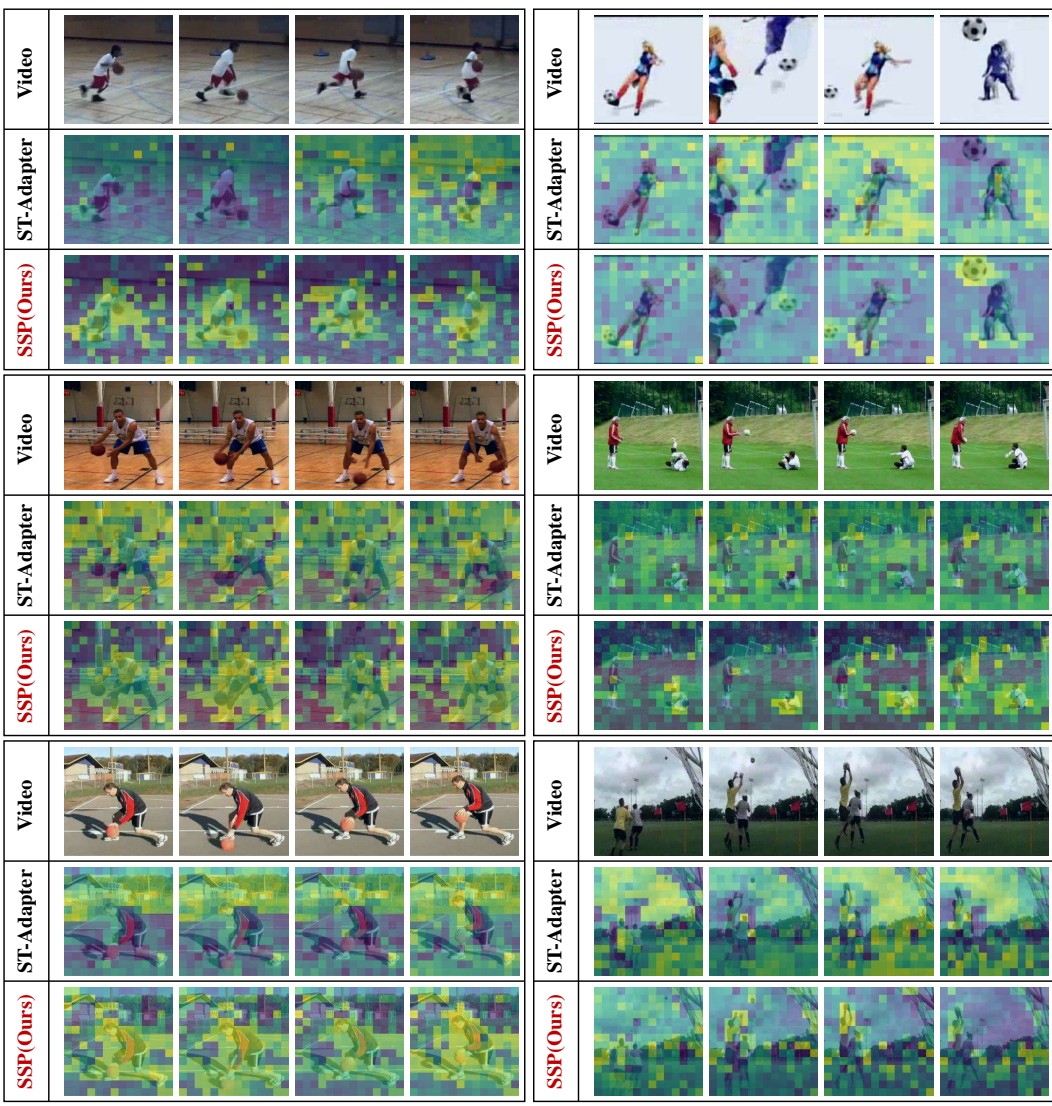

Figure 9: Visualization of our intra-frame prompts over the last layer in VideoMamba.

As shown in Figure 9, when existing parameter-efficient fine-tuning methods (e.g., ST-Adapter) are applied to VideoMamba, they fail to effectively capture the local feature maps of individual frames,

preventing the model from attending to local features that undergo temporal variations. In contrast, our approach, through efficient spreading of global temporal information, enables intra-frame prompts to more accurately capture discriminative local features, thereby achieving effective gathering of spatially relevant information.

## B.3 Inter-Frame Prompts Visualization

The inter-frame prompts in our SSP method facilitate the propagation of temporal information across sequential frames. Through visualization of these inter-frame prompts, we can identify which specific frames receive heightened attention from the model during global context integration, providing insights into the temporal dynamics of information processing within the architecture.

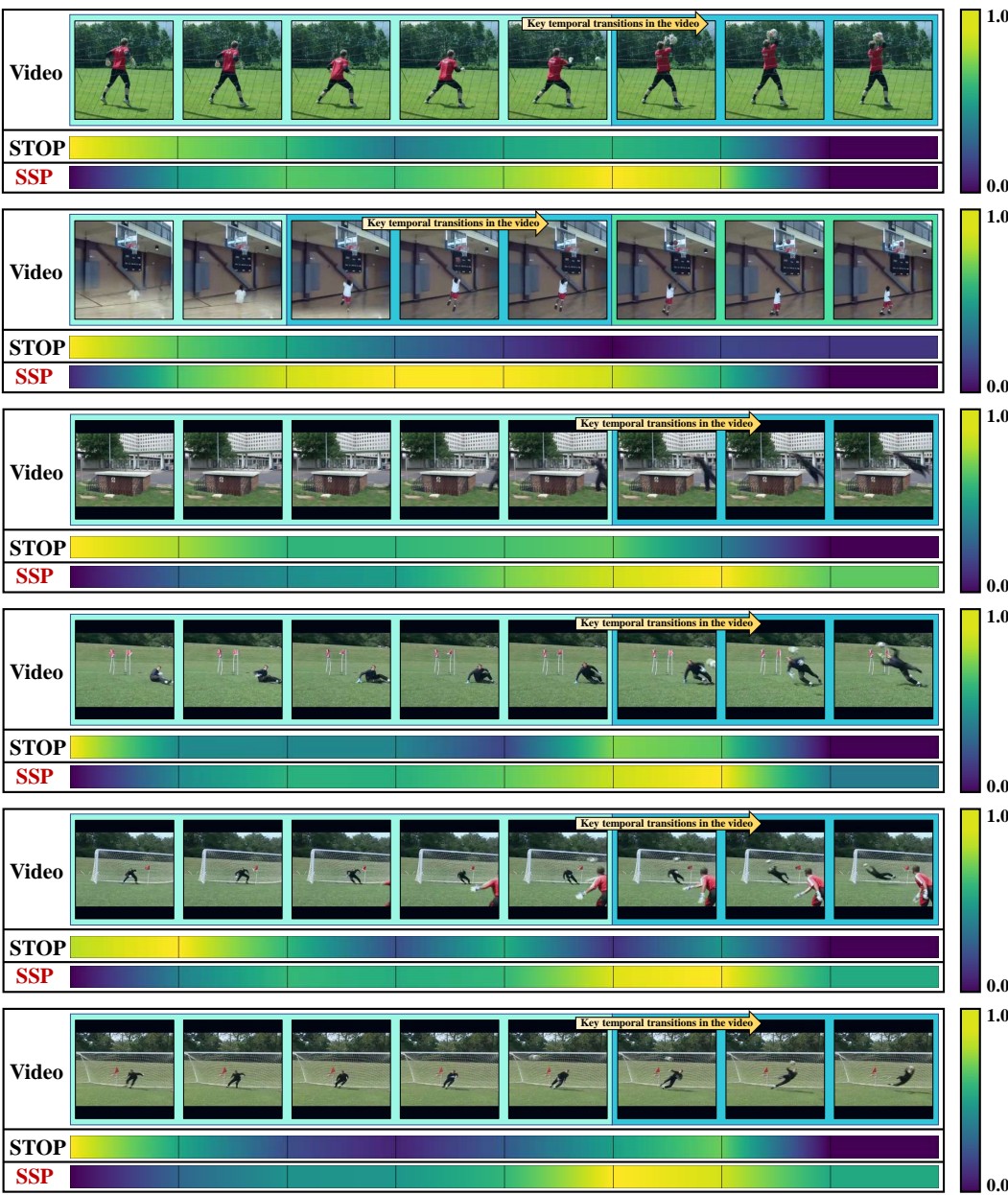

Figure 10: Visualization of our inter-frame prompts over the last layer in VideoMamba.

As demonstrated in Figure 10, compared to existing video prompting methods (e.g., STOP), the inter-frame prompts of our SSP method effectively identify key frames exhibiting temporal variations

within the video sequence. This enhanced capability stems from our approach being specifically engineered for the Mamba architecture, enabling efficient refinement and propagation of temporal information from compressed hidden states.

## C   Discussion on Different Method Designs

In our methodology design, we generate inter-frame prompts by sampling the last token from each frame in the forward scanning sequence, as illustrated in Figure 11(a). This section explores how different sampling strategies affect the generation of inter-frame prompts, with supplementary experimental results presented in Table 4. We discovered that sampling tokens

Table 4: Ablation of Different Sample Methods.

| Method | HMDB51 | UCF101 | Breakfast |
|---|---|---|---|
| Middle | 74.44 | 95.87 | 91.14 |
| Bidirection | 75.81 | 96.00 | 89.06 |
| Bi-Independent | 75.16 | 96.06 | 90.10 |
| SSP(Ours) | **76.66** | **97.03** | **93.23** |

from the middle of each frame (Figure 11(b)) leads to an average accuracy decrease of 1.82%. We attribute this decline to the fragmentation of semantic information within each frame during prompt generation. When sampling the last tokens separately from the forward and backward scanning sequences to generate prompts (Figure 11(c)), the accuracy decreases by an average of 2.01%. We posit that sampling before the superposition of bidirectional sequences causes the inter-frame prompts to overlook complementary information from both directions, resulting in a separation of forward and backward contextual cues. Additionally, employing independent inter-frame spreading modules to generate prompts for forward and backward directions (Figure 11(d)) still results in an accuracy drop of 1.86%. Although this approach introduces more learnable parameters to separately model contextual relationships in forward and backward scanning, it fails to address the fundamental issue of separated forward and backward contextual cues.

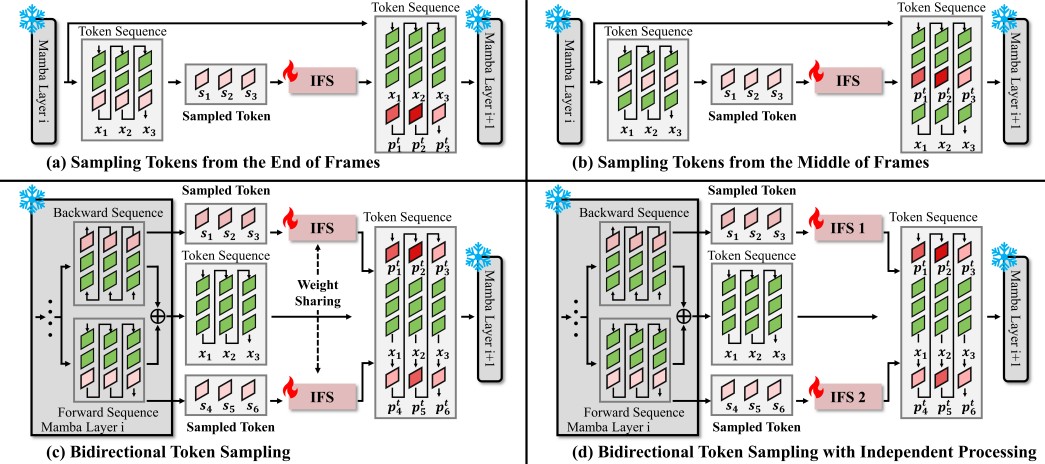

Figure 11: Different methods for inter-frame prompting. For simplicity, irrelevant computational processes in the Mamba block and intra-frame gathering modules have been omitted.

## D   More Ablation Studies on Hyperparameters

The internal dimensions of our intra-frame gathering module (IFG) and inter-frame spreading module (IFS) are also hyper-parameters of our method. To balance efficiency and effectiveness, we set the internal dimension of IFG to 384 and the internal dimension of IFS to 256. We conducted extensive experiments on HMDB51 and Breakfast datasets to investigate the impact of different dimension settings on fine-tuning performance.

As shown in Figure 12, naively increasing the internal dimension of the IFG module does not improve model performance. When the dimension is too large, model performance actually decreases due to increased optimization difficulty. Regarding the internal dimension of the IFS module, the long-video

Breakfast dataset is more sensitive to this setting. From the perspective of reducing training time and parameter costs, setting the internal dimension of the IFS to 32 is also an acceptable option, which demonstrates the robustness of our method to the choices of hyperparameters.

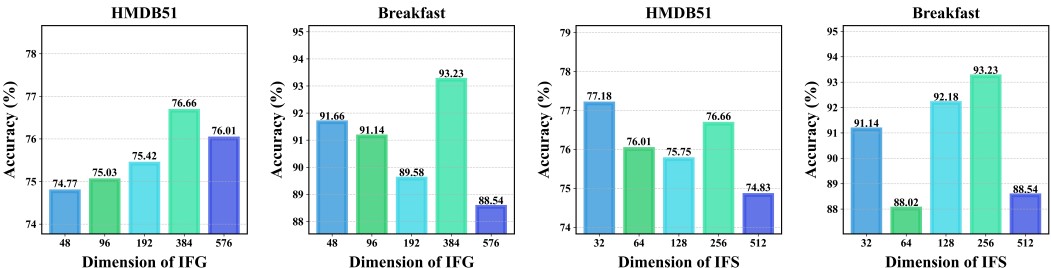

Figure 12: Ablation on the Internal Dimensions of IFG and IFS.

## E   More Clarifications

We first clarify the role of spatial variance in our approach. Given that different video frames exhibit varying degrees of association with global information, we learn the spatial variance $v_i$ through the spatial features aggregated by IFG. The spatial variance gates the weight of each inter-frame prompt, thereby controlling the degree of influence each video frame has on global information and achieving more refined temporal information propagation.

To further elaborate on the spreading mechanism, we describe how IFG and IFS work collaboratively. The IFG module aggregates spatial information from each frame into intra-frame prompts and superimposes the intra-frame prompts onto tokens. While the IFS module performs global attention computation through sampled tokens, enabling global interaction of the aggregated spatial information to generate inter-frame prompts. The inter-frame prompt corresponding to each frame thus contains spatial information from other frames, thereby propagating the aggregated local spatial information in a temporal manner.

## F   Asset License and Consent

DGL, VPT and VFPT are licensed under CC-BY-NC 4.0. Adapter, ST-Adapter, and CLIP-ViT are licensed under MIT. STOP, Additional-Scan and the VideoMamba are licensed under Apache 2.0.

All the datasets included in our study are publicly available, and all the models are publicly available. We would like to state that the contents in the dataset do NOT represent our views or opinions.

## G   Broader Impacts

This study presents SSP, which improves the performance of pre-trained Mamba models when fine-tuned on downstream video tasks. Thanks to the reduced parameter count, our research enables deployment of video foundation models on resource-constrained devices, reduced environmental impact of AI training, and rapid adaptation to specialized domains.

## H   Limitations

For potential limitations, our method introduces Intra-Frame Gathering (IFG) module and Inter-Frame Spreading (IFS) module to facilitate spatio-temporal contextual information, which brings additional hyperparameters, such as the number of IFSs. However, according to extensive experiments in § 4.5.3, we observe that in most cases, simply setting this parameter to 3 is sufficient. Regarding the internal dimensions of IFG and IFS, as shown in the extensive experiments in appendix D, we recommend setting the internal dimension of IFG to 384 and the internal dimension of IFS to 256.

