# OpenReview forum: "State Space Prompting via Gathering and Spreading Spatio-Temporal Information for Video Understanding"
_NeurIPS.cc/2025/Conference — NeurIPS 2025 poster_

### Official Review · Reviewer_Ab59 · 2025-06-20

**Clarity:** 2
**Significance:** 2
**Originality:** 2
**Rating:** 4
**Confidence:** 3

**Summary:**

Pre-trained state space models compress visual tokens efficiently for video classification but struggle to capture spatial and temporal context. To address this, the authors propose State Space Prompting (SSP), which uses intra-frame and inter-frame prompts to better gather and spread key spatiotemporal information. The Intra-Frame Gathering module captures spatial details within frames, while the Inter-Frame Spreading module shares important cues across frames. SSP improves video understanding by effectively balancing and compressing this information. Experiments conducted on four video benchmark datasets demonstrate the contribution of the approach.

**Questions:**

1) What is the specific technical novelty of the proposed approach beyond incremental improvements, and how does it effectively learn discriminative spatio-temporal information?

2) How exactly do the Intra-Frame Gathering and Inter-Frame Spreading modules spread gathered local spatial information in a temporal manner?

3) Can the authors clearly explain how the proposed spatial variant works and how it enhances the representation of discriminative action features?

**Ethical Concerns:**

["NO or VERY MINOR ethics concerns only"]

**Final Justification:**

Considering the clarifications provided by the authors during the rebuttal period, some of my concerns and questions have been addressed. I am inclined to increase my score to borderline accept

**Limitations:**

yes

**Paper Formatting Concerns:**

No problem.

**Quality:**

2

**Strengths And Weaknesses:**

The motivation of the proposed approach makes sense, as it enhances the understanding of spatiotemporal information through intra-frame and inter-frame prompts. It is easy to read and follow. The proposed approach achieves state-of-the-art results, as shown in Table 1 and Table 2.

While the proposed approach shows some technical improvement, its novelty is limited and largely incremental. It remains unclear how the method effectively learns discriminative spatio-temporal features. Additionally, the explanation in lines 60–65 regarding the “Intra-Frame Gathering module and Inter-Frame Spreading module facilitating spatio-temporal contextual interaction by spreading gathered local spatial information temporally” is vague and lacks clarity on the actual mechanism of temporal spreading.

Moreover, the presentation must clearly explain how the proposed spatial variant operates and how it enhances the representation of discriminative action features. Without this, the contribution to improving spatiotemporal feature learning is not convincingly demonstrated.

It is not clear how the parameters are calculated. Figure 5 shows 1.30M parameters, but the numbers in Table 1 and Table 2 are different.

---

> ### Author Rebuttal · Authors · 2025-07-30
>
> ### W1 & Q1 Response: What is the specific technical novelty of the proposed approach beyond incremental improvements, and how does it effectively learn discriminative spatio-temporal information? What is the actual mechanism of "spreading gathered local spatial information temporally"?
>
> Thank you for your valuable review. **We highlight the following aspects to illustrate the novelty and operational mechanism of our approach:**
>
> (1) **Firstly, unlike existing fine-tuning methods, our approach is specifically designed for Mamba architecture models.**
>
> - Through the low-rank convolution of the IFG module, **we learn discriminative spatial features and superimpose the aggregated local spatial information onto tokens**. The IFG module effectively introduces two-dimensional spatial inductive bias into sequential Mamba models while requiring only **~3%**(**2.41M/74.00M**) of tunable parameters compared to full fine-tuning, as shown in Table 2, enabling effective spatial feature aggregation and learning.
>
> - Leveraging the inherent characteristic of Mamba models to sequentially compress tokens, our IFS generates inter-frame prompts through global attention using sampled tokens from each frame, **achieving temporal propagation of local spatial information in this process**. As discussed in Sec 6.1 of the appendix, the IFS module significantly reduces the path length for information transmission, facilitating the flow of temporal information.
>
> (2) **Secondly, the proposed IFG and IFS work in a complementary manner to learn discriminative spatio-temporal information.**
>
> - As mentioned above, the intra-frame prompts generated by IFG are superimposed onto the tokens, enabling intra-frame tokens to acquire discriminative local spatial information.
>
> - IFS conducts low-rank attention computation on sampled tokens, allowing the aggregated spatial information from each frame to interact and generate inter-frame prompts.
>
> - The inter-frame prompts are inserted at the boundaries between frames and fed into the next Mamba layer, enabling the Mamba layer to simultaneously model the local spatial information contained in intra-frame tokens and the global temporal information contained in inter-frame prompts when processing tokens.
>
> (3) **We clarify the spreading mechanism of gathered local spatial information through the IFG and IFS modules as follows:**
>
> - **Firstly**, the IFG module **aggregates spatial information from each frame** into intra-frame prompts and superimposes the intra-frame prompts onto tokens. **Secondly**, the IFS module performs global attention computation through sampled tokens, **enabling global interaction of the aggregated spatial information** to generate inter-frame prompts. **Thirdly**, the inter-frame prompt corresponding to each frame thus contains spatial information from other frames, **thereby propagating the aggregated local spatial information in a temporal manner**.
>
> - As shown by the visualization study in Figure 9 in the appendix, the intra-frame prompts generated by the IFG module **successfully aggregate key local spatial information.** The intra-frame prompts are superimposed onto tokens, further facilitating the IFS in propagating local spatial information.
>
> - As demonstrated by the visualization study in Figure 10 in the appendix, after the propagation of discriminative information in a temporal manner, **our method successfully focuses on key temporal transitions in the video,** which further verifies our effectiveness.
>
> - As shown by the ablation experiments in Table 3, when the IFG module is removed, the accuracy decreases by an average of **4.25%**, and when the IFS module is removed, the accuracy decreases by an average of **3.67%**. **This indicates that the IFG and IFS modules play important roles in the aggregation and propagation of discriminative spatiotemporal information.**
>
> (4) As shown in Table 1 and Table 2, our method achieves average improvements of **1.21%** and **2.76%** compared to existing state-of-the-art fine-tuning methods, realizing effective learning of discriminative spatiotemporal features, which further verifies our effectiveness.
>
> Table 1. The comparison results on VideoMamba-S (Parameters **25.42M**)
>
> | Method | Param | HMDB51 | SSV2 | UCF101 | Breakfast | Avg |
> |--------|-------|--------|------|--------|-----------|-----------|
> | Full | 25.42M | 67.58 | 58.57 | 92.96 | 94.27 | 78.34 |
> | Adapter | 2.40M | *73.79* | 36.45 | 94.18 | *84.89* | *72.33* |
> | ST-Adapter | 2.69M | 70.52 | 30.94 | 94.87 | 77.60 | 68.48 |
> | VPT | 1.50M | 72.74 | 30.68 | 95.16 | 81.25 | 69.95 |
> | VFPT | 1.50M | 72.41 | 30.37 | 95.08 | 79.68 | 69.38 |
> | SVP | 2.76M | 69.93 | *38.01* | 95.58 | 80.72 | 71.06 |
> | Additional-Scan | 0.66M | 73.20 | 33.71 | *95.63* | 78.64 | 70.29 |
> | STOP | 1.49M | 70.06 | 21.22 | 93.44 | 65.62 | 62.58 |
> | **SSP** *(Ours)* | 0.98M | **74.38(+0.59)** | **38.68(+0.67)** | **95.69(+0.06)** | **85.41(+0.52)** | **73.54(+1.21)** |
>
> Table 2. The comparison results on VideoMamba-M (Parameters **74.00M**)
>
> | Method | Param | HMDB51 | SSV2 | UCF101 | Breakfast | Avg |
> |--------|-------|--------|------|--------|-----------|-----|
> | Full | 74.00M | 76.30 | 67.30 | 96.00 | 95.31 | 83.73 |
> | Adapter | 2.40M | 73.59 | 46.98 | 96.24 | 89.06 | 76.47 |
> | ST-Adapter | 2.69M | 71.96 | 31.91 | 94.81 | 67.70 | 66.60 |
> | VPT | 1.50M | 73.59 | 40.58 | 95.45 | 81.77 | 72.85 |
> | VFPT | 1.50M | 71.89 | 39.68 | 95.71 | 86.97 | 73.56 |
> | SVP | 2.76M | *73.66* | *49.08* | *96.77* | *90.10* | *77.40* |
> | Additional-Scan | 1.33M | 73.52 | 44.65 | 96.32 | 86.97 | 75.37 |
> | STOP | 1.49M | 71.96 | 23.44 | 94.60 | 71.35 | 65.34 |
> | **SSP** *(Ours)* | 2.41M | **76.66(+3.00)** | **53.72(+4.64)** | **97.03(+0.26)** | **93.23(+3.13)** | **80.16(+2.76)** |
>
> Table 3. Ablation of different prompting modules
>
> | IFG | IFS | HMDB51 | UCF101 | Breakfast | Avg |
> |-----|-----|--------|--------|-----------| -------- |
> | - | - | 59.34(-17.32) | 90.64(-6.39) | 76.56(-16.67) | 75.51(-13.46) |
> | √ | - | 74.44(-2.22) | 96.56(-0.47) | 84.89(-8.34) | 85.30(-3.67) |
> | - | √ | 72.61(-4.05) | 96.14(-0.89) | 85.41(-7.82) | 84.72(-4.25) |
> | √ | √ | **76.66** | **97.03** | **93.23** | **88.97** |
>
> ### W2 & Q3 Response: Can the authors clearly explain how the proposed spatial variant works and how it enhances the representation of discriminative action features?
>
> Thanks for your valuable question. We elucidate the working mechanism of spatial variance and its approach to enhancing discriminative action feature representation through the following three points:
>
> (1) As shown in Figure 2 in the main text, the inter-frame prompts generated by IFS correspond one-to-one with the video frames from which the sampled tokens originate, and they are inserted at the boundaries between video frames, **enabling the next Mamba layer to simultaneously model the global temporal information brought by inter-frame prompts when processing adjacent frames.**
>
> (2) However, different video frames have varying degrees of association with global information. We aim to further learn a feature called "spatial variant" through the spatial features aggregated by IFG, where spatial variant gates the weight of each inter-frame prompt, **thereby controlling the degree of influence each video frame has on global information and achieving more refined temporal information propagation.**
>
> (3) As shown in the ablation experiment results in Table 4, when the spatial variant mechanism is removed, the classification accuracy drops by an average of **2.33%**, indicating that **the spatial variant mechanism is beneficial for enhancing the modeling capability of our method.**
>
> In summary, spatial variant can achieve more refined temporal information propagation and enhance the representation of discriminative action features **by effectively controlling the influence of local information on global information**.
>
> Table 4. Ablation of spatial variant
>
> | Entropy | Spatial | HMDB51 | UCF101 | Breakfast | Avg |
> |---------|---------|--------|--------|-----------| -------- |
> | - | - | 74.70(-1.96) | 96.56(-0.47) | 89.58(-3.65) | 86.95(-2.02) |
> | √ | - | 75.09(-1.57) | 96.29(-0.74) | 88.54(-4.69) | 86.64(-2.33) |
> | - | √ | 74.96(-1.70) | 96.61(-0.42) | 88.54(-4.69) | 86.70(-2.27) |
> | √ | √ | **76.66** | **97.03** | **93.23** | **88.97** |
>
> ### W3 Response: How the parameters are calculated? Figure 5 shows 1.30M parameters, but the numbers in Table 1 and Table 2 are different.
>
> The parameter variation in our method stems from changes in the number of IFS modules, **which is demonstrated by the ablation experiment in Figure 7 in the main text.**
>
> The **1.30M** parameters shown in Figure 5 come from a configuration using only one IFS module, intended to demonstrate that our method surpasses the performance of existing methods while using the minimum number of parameters. This is explained in both the caption of Figure 5 and lines **273-274** of the main text. **Our method achieves better performance under the same parameter scale, which further verifies our effectiveness**.
>
>
> ### Q2 Response: How exactly do the Intra-Frame Gathering and Inter-Frame Spreading modules spread gathered local spatial information in a temporal manner?
>
> Thanks for your valuable question. **We have clarified the spreading mechanism of gathered local spatial information in the W1 & Q1 section above.**
>
> **Firstly**, the IFG module aggregates spatial information from each frame. **Secondly**, the IFS module enables global interaction of the aggregated spatial information. **Thirdly**, the inter-frame prompt corresponding to each frame thus contains spatial information from other frames, propagating the aggregated information in a temporal manner.
>
> As shown in Table 4, when the IFG or IFS module is removed, the accuracy decreases by an average of **4.25%** or **3.67%**. **This indicates the important roles of IFG and IFS modules in the aggregation and propagation of spatiotemporal information.**

---

> > ### Comment · Reviewer_Ab59 · 2025-08-05
> >
> > Thank you to the authors for their good response. After reviewing the authors' rebuttal and the feedback from the other reviewer, as well as my previous comments, I find the motivation for the proposed approach well justified.
> >
> > 1. About novelty.   My concerns about novelty have been addressed satisfactorily. I will adjust my score based on the authors' response and the other reviewers' comments.
> >
> > 2. About IFG and IFS Clarity. The authors' response concerning the IFG and IFS modules is satisfactory. I request the authors that they provide and include a clearer explanation in the updated manuscript.
> >
> > 3. About Spreading Mechanism: Thank you for the detailed response regarding the spreading mechanism of gathered local spatial information. I appreciate the authors' effort. Please also consider adding a clearer explanation in the revised manuscript.
> >
> > 4. Spatial Variant Works and Parameters: The authors' response  is satisfies my concerns.
> >
> > Additionally, I noted the concern raised by Reviewer VAWP: "The IFG and IFS modules perform heavy information aggregation, which may lead to a loss of fine-grained spatial or temporal details."
> >
> > Overall, I have carefully reviewed the authors' responses and the discussion among reviewers. Considering the clarifications provided, I am inclined to increase my score to borderline accept. Thank you to the authors for their detailed and constructive responses.

---

> > > ### Author Response · Authors · 2025-08-06
> > >
> > > Dear Reviewer Ab59,
> > >
> > > Thank you for your thoughtful feedback and for taking the time to carefully review our rebuttal. We are very grateful for your updated score and appreciate your positive assessment of our work.
> > >
> > > We will certainly incorporate your valuable suggestions into our revised manuscript. Specifically, we will add clearer explanations of the IFG and IFS modules, the spreading mechanism, and the mechanism of our Spatial Variant to improve the overall clarity of the paper. We believe these changes will significantly strengthen our work.
> > >
> > > Thank you once again for your constructive comments and support.
> > >
> > > Sincerely,
> > >
> > > The Authors of Paper 3493

---

> > > > ### Comment · Reviewer_Ab59 · 2025-08-07
> > > >
> > > > As mentioned by Reviewer VAWP: both the IFG and IFS modules perform heavy compression through low-rank operations, which may result in the loss of fine-grained spatial details. Could you please provide more details and clarification about this?
> > > >
> > > > Thank you.

---

> > > > > ### Author Response · Authors · 2025-08-07
> > > > > **More Details and Clarification about Fine-grained Spatial Details**
> > > > >
> > > > > Thank you for your valuable question. **Our IFG and IFS modules both perform low-rank projection during convolution and attention computation, which effectively extract and propagate discriminative spatial information while preserving fine-grained spatial details.** We conducted ablation experiments, visualization study and additional experiments on spatially demanding task to verify the actual spatial modeling capabilities of our proposed method.
> > > > >
> > > > > (1) **We conducted ablation experiments to demonstrate the effectiveness of low-rank operations in our IFG and IFS modules.** As shown in Table 1 and Table 2, on the HMDB51 and Breakfast datasets, when the internal dimension of IFG is increased to **576**, the accuracy decreases by **0.65%** and **4.69%** compared to the optimal configuration, respectfully. When the internal dimension of IFS is increased to **512**, the accuracy decreases by **2.35%** and **4.69%**, respectfully. Therefore, the low-rank operations in our IFG and IFS modules are effective and necessary to ensure effective aggregation of spatiotemporal details.
> > > > >
> > > > > (2) As shown by the visualization study in Figure 9 in the appendix, **our intra-frame prompts can preserve fine-grained spatial details compared to other fine-tuning methods.** Through effective extraction of fine-grained spatial information, our intra-frame prompts can better capture discriminative spatial features, which further validates the effectiveness of our method.
> > > > >
> > > > > (3) **We have also conducted additional experiments on video object segmentation task, which is spatially demanding.** We utilized the VTUS[1] dataset from the medical video segmentation domain, which exhibits high sensitivity to spatial details, to comprehensively evaluate our method's spatial modeling capabilities.
> > > > >
> > > > > As shown in Table 3, our SSP outperforms existing state-of-the-art fine-tuning methods by **1.65%**, **2.01%**, **1.22%**, and **0.55%** on the Dice, Jaccard, Precision, and Recall metrics, respectively. Our method's superior fine-tuning segmentation performance comes from the IFG and IFS modules, **which use efficient low-rank convolutions and attention to extract key spatial information and propagate it while preserving fine-grained details.** These results further validate our method's effectiveness at extracting fine-grained spatial details.
> > > > >
> > > > > To summarize, our method shows superior spatial modeling compared to current state-of-the-art fine-tuning approaches for Mamba architectures. The visualization study and additional experiments on video object segmentation task provide stronger support for our claim of improved spatial modeling capabilities. As shown in our previous response, the clarification we provided has addressed most of the concerns of reviewer VAWP, who initially raised this issue. We will continue to track relevant research developments and explore the application of our method to other spatially demanding tasks in future investigations.
> > > > >
> > > > > Table 1. Ablation on the Internal Dimensions of IFG
> > > > >
> > > > > | Methods \ Dimensions | 48    | 96    | 192   | 384   | 576   |
> > > > > |---------|-------|-------|-------|-------|-------|
> > > > > | HMDB51  | 74.77(-1.89) | 75.03(-1.63) | 75.42(-1.24) | **76.66** | 76.01(-0.65) |
> > > > > | Breakfast | 91.66(-1.57) | 91.14(-2.09) | 89.58(-3.65) | **93.23** | 88.54(-4.69) |
> > > > >
> > > > > Table 2. Ablation on the Internal Dimensions of IFS
> > > > >
> > > > > | Methods \ Dimensions | 32    | 64    | 128   | 256   | 512   |
> > > > > |---------|-------|-------|-------|-------|-------|
> > > > > | HMDB51  | **77.18** | 76.01(-1.17) | 75.75(-1.43) | 76.66(-0.52) | 74.83(-2.35) |
> > > > > | Breakfast | 91.14(-2.09) | 88.02(-5.21) | 92.18(-1.05) | **93.23** | 88.54(-4.69) |
> > > > >
> > > > > Table 3. Comparison on the VTUS dataset.
> > > > >
> > > > > | Method | Backbone | Param | Dice | Jaccard | Precision | Recall |
> > > > > |--------|--------|-------|------|---------|-----------|--------|
> > > > > | VPT | Vivim | 0.81M | 0.6670 | 0.5444 | *0.6721* | 0.8247 |
> > > > > | VFPT | Vivim | 0.81M | 0.6657 | 0.5435 | 0.6681 | 0.8279 |
> > > > > | SVP | Vivim | 1.54M | *0.6708* | *0.5495* | 0.6709 | *0.8365* |
> > > > > | **SSP** *(Ours)* | Vivim | 0.97M | **0.6873(+0.0165)** | **0.5696(+0.0201)** | **0.6843(+0.0122)** | **0.8420(+0.0055)** |
> > > > >
> > > > >
> > > > >
> > > > > [1] Yang, Yijun, et al. "Vivim: A video vision mamba for medical video segmentation." arXiv preprint arXiv:2401.14168 (2024).

---

### Official Review · Reviewer_VAWP · 2025-06-29

**Clarity:** 2
**Significance:** 1
**Originality:** 1
**Rating:** 3
**Confidence:** 4

**Summary:**

This paper proposes State Space Prompting (SSP), a parameter-efficient fine-tuning method for VideoMamba, a state space model for video understanding. SSP introduces two lightweight modules—Intra-Frame Gathering (IFG) and Inter-Frame Spreading (IFS)—to capture spatial cues within frames and propagate temporal information across frames, mitigating information decay in state space compression. SSP enables efficient adaptation to downstream video tasks with minimal tunable parameters and achieves state-of-the-art results across multiple video benchmarks.

**Questions:**

1. Have you evaluated the proposed method on more complex video understanding tasks beyond classification, such as temporal action segmentation or video object segmentation?
Without such evaluations, it is unclear whether the IFG and IFS modules truly benefit spatio-temporal reasoning as claimed.

How do you mitigate the risk of losing fine-grained spatial or temporal details due to the heavy aggregation performed by IFG and IFS?
Have any ablation or visualization studies been conducted to confirm that key local information is preserved?

Given that the method is only applied to video classification, would you consider narrowing the paper’s scope or revising the title to better reflect its actual contributions?
This would help manage reader expectations and align the paper's claims with its experiments.

**Ethical Concerns:**

["NO or VERY MINOR ethics concerns only"]

**Final Justification:**

The authors' rebuttal has addressed most of my concerns. I am currently considering raising my score and will make a final decision during the discussion phase.

**Limitations:**

yes

**Paper Formatting Concerns:**

The authors appear to have followed the NeurIPS 2025 Paper Formatting Instructions correctly

**Quality:**

2

**Strengths And Weaknesses:**

Strengths
The paper is technically sound and proposes a novel architecture that integrates state space modeling with spatio-temporal feature manipulation via Information Gathering (IFG) and Information Spreading (IFS) modules. The implementation appears well-executed, and the model achieves solid results on several standard video classification benchmarks. The paper is clearly written and well-structured. The roles of IFG and IFS are well explained, and diagrams help convey the pipeline effectively.

Weaknesses
1. Despite the title claiming a contribution to "video understanding," the paper only evaluates on video classification, which is relatively simple and insufficient to validate the claimed benefits of IFG and IFS for spatio-temporal reasoning. It would be more convincing to test on video segmentation (e.g., instance/object/panoptic) or temporal tasks like action segmentation, detection, or localization.

2. The IFG and IFS modules perform heavy information aggregation, which may lead to a loss of fine-grained spatial or temporal details. This could limit the method's effectiveness on tasks that require precise local reasoning, especially outside of classification.

3. If the proposed approach is designed specifically for video classification, the title should reflect that scope more accurately. A clearer title would be:
"State Space Prompting via Gathering and Spreading Spatio-Temporal Information for Video Classification.

---

> ### Author Rebuttal · Authors · 2025-07-30
>
> ### W1 & Q1 Response: Evaluation on more tasks beyond video classification
>
> Thank you for your valuable suggestion. **We have conducted more experiments on video understanding tasks beyond video classification.**
>
> Following the parameter-efficient video learning community, we conducted fair comparisons with existing fine-tuning methods under the same video classification paradigm using four mainstream datasets in the main text. **Additionally, we also compared with existing methods on the THUMOS dataset under the temporal action localization paradigm and on the 50salads and GTEA datasets under the temporal action segmentation paradigm.**
>
> (1) For the temporal action localization paradigm, as shown in Table 1, the average mAP score of our method exceeds existing state-of-the-art fine-tuning methods by **1.83**. Our proposed SSP demonstrates stronger temporal action localization capability through efficient aggregation and propagation of discriminative spatiotemporal features, which further showcases the effectiveness of SSP in challenging spatial-temporal reasoning tasks.
>
> (2) For the temporal action segmentation paradigm, as shown in Table 2, on the 50salads dataset, SSP achieves an accuracy that exceeds the existing best fine-tuning method by **2.90%**, with the F1@50 score obtaining an improvement of **2.27**. As shown in Table 3, on the GTEA dataset, SSP achieves an accuracy that surpasses existing state-of-the-art methods by **1.47%**, with the F1@50 score also obtaining an improvement of **1.73**. This demonstrates that our method achieves efficient extraction of discriminative spatiotemporal information from Mamba hidden states through effective aggregation of local information and efficient propagation of temporal information, **making improvements in spatio-temporal reasoning on tasks beyond classification as well**, which further verifies the effectiveness of our SSP method.
>
> **In summary, all the above experimental results demonstrate our effectiveness and generalization ability on various video tasks.**
>
> Table 1. Comparison on the THUMOS datasets under the temporal action localization paradigm
>
> | Methods | Backbone     | Params | mAP@0.3 | mAP@0.4 | mAP@0.5 | mAP@0.6 | mAP@0.7 | mAP@Avg |
> |---------|--------------|--------|---------|---------|---------|---------|---------|---------|
> | Adapter | ActionMamba  | 1.05M  | *81.87*   | *77.37*   | *69.67*   | *56.82*   | *41.36*   | *65.42*   |
> | SVP     | ActionMamba  | 1.31M  | 81.56   | 77.33   | 68.78   | 56.65   | 38.21   | 64.51   |
> | **SSP** *(Ours)*    | ActionMamba  | 1.05M  | **83.40(+1.53)**   | **78.88(+1.51)**   | **71.40(+1.73)**   | **59.55(+2.73)**   | **43.00(+1.64)**   | **67.25(+1.83)**   |
>
> Table 2. Comparison on the 50salads datasets under the temporal action segmentation paradigm
>
> | Methods | Backbone | Params | Acc   | Edit  | F1@10 | F1@25 | F1@50 |
> |---------|----------|--------|-------|-------|-------|-------|-------|
> | Adapter | ASMamba  | 0.82M  | 70.45 | 51.25 | 57.62 | 53.38 | 43.64 |
> | VPT     | ASMamba  | 0.49M  | *72.75* | 50.92 | 59.70 | 57.14 | *45.20* |
> | VFPT    | ASMamba  | 0.49M  | 69.74 | **57.98** | *61.02* | *58.35* | 44.09 |
> | SVP     | ASMamba  | 0.88M  | 68.63 | 51.68 | 56.75 | 53.08 | 42.56 |
> | **SSP** *(Ours)* | ASMamba  | 0.82M  | **75.65(+2.90)** | *53.58* | **64.61(+3.59)** | **60.65(+2.30)** | **47.47(+2.27)** |
>
> Table 3. Comparison on the GTEA datasets under the temporal action segmentation paradigm
>
> | Methods | Backbone | Params | Acc   | Edit  | F1@10 | F1@25 | F1@50 |
> |---------|----------|--------|-------|-------|-------|-------|-------|
> | Adapter | ASMamba  | 0.81M  | 66.71 | 64.85 | 65.88 | 58.82 | 40.00 |
> | VPT     | ASMamba  | 0.48M  | 66.69 | *66.54* | *67.20* | 60.80 | *44.80* |
> | VFPT    | ASMamba  | 0.48M  | 65.87 | 65.01 | 66.12 | *61.22* | 39.18 |
> | SVP     | ASMamba  | 0.87M  | *67.39* | 66.49 | 65.30 | 60.40 | 40.81 |
> | **SSP** *(Ours)* | ASMamba  | 0.81M  | **68.86(+1.47)** | **73.91(+7.37)** | **71.83(+4.63)** | **66.12(+4.90)** | **46.53(+1.73)** |
>
> ### W2 Response: The potential loss of fine-grained spatial or temporal details
>
> **Compared to existing fine-tuning methods, our SSP method achieves better preservation of fine-grained spatial and temporal details.**
>
> Therefore, we performed further experiments on temporal action localization and temporal action segmentation tasks to validate the effectiveness of our approach in extracting fine-grained video spatiotemporal features.
>
> (1) Regarding the temporal action localization paradigm, the average mAP score of our method exceeds existing state-of-the-art fine-tuning methods by **1.83**, as shown in Table 1, demonstrating better fine-grained spatiotemporal understanding capability during fine-tuning. This is because our IFG and IFS modules both *perform low-rank projection during convolution and attention computation*, enabling the modules to effectively aggregate and propagate discriminative spatiotemporal information while preserving discriminative fine-grained spatial or temporal details.
>
> (2) Regarding the temporal action segmentation paradigm, our SSP surpasses existing fine-tuning methods with accuracy gains of **2.90%** and **1.47%** demonstrated in Table 2 and Table 3, respectively. It illustrates that *our approach better preserves fine-grained spatial and temporal details through the 2D local convolution of the IFG module and the inter-frame information propagation of the IFS module,* which further proves the effectiveness of our fine-tuning method on the Mamba architecture.
>
> In summary, these experimental results above indicate that our SSP method does not suffer from the potential loss of fine-grained spatial or temporal details. Instead, our approach successfully preserves and leverages fine-grained spatial and temporal details, validating the effectiveness of our proposed fine-tuning method.
>
> ### Q2 Response: How do you mitigate the risk of losing fine-grained spatial or temporal details due to the heavy aggregation performed by IFG and IFS?
>
> **In order to effectively aggregate and propagate fine-grained discriminative spatiotemporal information, our IFG and IFS modules both perform low-rank projection during convolution and attention computation.**
>
> (1) Regarding the temporal action localization paradigm, the average mAP score of our method exceeds the SOTA fine-tuning methods by **1.83**, as shown in Table 1, demonstrating better fine-grained spatiotemporal understanding capability. Regarding the temporal action segmentation paradigm, our SSP surpasses SOTA methods with accuracy gains of **2.90%** and **1.47%** demonstrated in Table 2 and Table 3, respectively. This is because our low rank IFG and IFS modules effectively aggregate and propagate discriminative information while preserving discriminative fine-grained spatial or temporal details, which further proves the effectiveness of our fine-tuning method.
>
> (2) We conducted ablation experiments to demonstrate the effectiveness of our approach. As shown in Table 4 and Table 5, on the HMDB51 and Breakfast datasets, when the internal dimension of IFG is increased to **576**, the accuracy decreases by **0.65%** and **4.69%** compared to the optimal configuration, respectfully. When the internal dimension of IFS is increased to **512**, the accuracy decreases by **2.35%** and **4.69%**, respectfully. Therefore, **we achieve a balance between heavy aggregation and fine-grained modeling** by selecting appropriate intrinsic low-rank dimensions for IFG and IFS to ensure effective aggregation of spatiotemporal details.
>
> (3) As shown by the visualization study in Figure 9 in the appendix, even after heavy aggregation, **our intra-frame prompts can still preserve fine-grained spatial details compared to other fine-tuning methods.** As demonstrated by the visualization study in Figure 10 in the appendix, **our intra-frame prompts accurately focus on key temporal transitions in the video, preserving fine-grained temporal information,** which further verifies our effectiveness.
>
> Table 4. Ablation on the Internal Dimensions of IFG
>
> | Methods \ Dimensions | 48    | 96    | 192   | 384   | 576   |
> |---------|-------|-------|-------|-------|-------|
> | HMDB51  | 74.77(-1.89) | 75.03(-1.63) | 75.42(-1.24) | **76.66** | 76.01(-0.65) |
> | Breakfast | 91.66(-1.57) | 91.14(-2.09) | 89.58(-3.65) | **93.23** | 88.54(-4.69) |
>
> Table 5. Ablation on the Internal Dimensions of IFS
>
> | Methods \ Dimensions | 32    | 64    | 128   | 256   | 512   |
> |---------|-------|-------|-------|-------|-------|
> | HMDB51  | **77.18** | 76.01(-1.17) | 75.75(-1.43) | 76.66(-0.52) | 74.83(-2.35) |
> | Breakfast | 91.14(-2.09) | 88.02(-5.21) | 92.18(-1.05) | **93.23** | 88.54(-4.69) |
>
> ### W3 & Q3 Response: About the title
>
> Thank you for your suggestion. We have already supplemented experiments on temporal action localization and temporal action segmentation in previous responses and validated the effectiveness of our method in fine-grained spatiotemporal feature extraction.
>
> As shown in Table 1, the average mAP score of our method exceeds existing state-of-the-art fine-tuning methods by **1.83** on the temporal action localization task. As shown in Table 2 and Table 3, our method achieves accuracy gains of **2.90%** and **1.47%** on the temporal action segmentation task, respectively. These results indicate that our SSP method effectively aggregates and propagates discriminative spatiotemporal information, proving our effectiveness and generalization ability on various video understanding tasks.

---

> > ### Comment · Reviewer_VAWP · 2025-08-01
> > **Request for More Evidence of Spatial Modeling Capabilities (e.g., VIS/VPS Tasks)**
> >
> > While we appreciate the thorough evaluations on classification, temporal action localization, and segmentation tasks, we still have concerns regarding the actual spatial modeling capabilities of the proposed SSP method.
> >
> > Both IFG and IFS modules perform heavy compression through low-rank operations, which may result in the loss of fine-grained spatial details. Although you provided indirect evidence via visualization and accuracy gains on coarse-grained tasks like video classification or temporal action segmentation, these tasks are not necessarily sensitive to spatial details.
> >
> > To further validate your claim that SSP effectively preserves and propagates spatial information, it would be more convincing to include evaluations on spatially demanding tasks such as video instance segmentation (VIS) or video panoptic segmentation (VPS). These tasks directly measure the model's ability to localize and differentiate fine-grained visual regions over time.
> >
> > Could you consider adding or commenting on such evaluations to better support your claim of improved spatial modeling?

---

> > > ### Author Response · Authors · 2025-08-03
> > > **More Experiments on Video Segmentation Task**
> > >
> > > Thank you for your valuable question. **We have conducted additional experiments on spatially demanding task to verify the actual spatial modeling capabilities of our proposed method.**
> > >
> > > (1) Regarding **video instance segmentation (VIS) and video panoptic segmentation (VPS) tasks**, although we are eager to further validate our method's spatial modeling capabilities on these tasks, **we have not identified any open-source VIS or VPS backbone models built upon the Mamba architecture**.
> > >
> > > The core contribution of our approach lies in enabling efficient fine-tuning of Mamba-based models through effective extraction and propagation of spatiotemporal information from Mamba hidden states. Consequently, applying our method to **Mamba-based backbone models for VIS or VPS** would yield more significant insights.
> > >
> > > We would greatly appreciate any guidance you could provide regarding existing open-source works that have proposed Mamba-based backbone architectures for VIS or VPS tasks.
> > >
> > > (2) Regarding the **video object segmentation task** which you have mentioned in Weakness 1 and Question 1, we have conducted additional experiments to address this concern. We utilized the VTUS[1] dataset from the medical video segmentation domain, which exhibits high sensitivity to spatial details, to comprehensively evaluate our method's spatial modeling capabilities.
> > >
> > > As demonstrated in Table 1, our SSP outperforms existing state-of-the-art fine-tuning methods by **1.65%**, **2.01%**, **1.22%**, and **0.55%** on the Dice, Jaccard, Precision, and Recall metrics, respectively. This improvement stems from our IFG and IFS modules, which **leverage parameter-efficient low-rank convolutions and attention mechanisms to effectively extract and propagate discriminative spatial information while preserving fine-grained spatial details**, consequently achieving superior segmentation performance during fine-tuning. These results further validate the effectiveness of our method in spatially demanding tasks.
> > >
> > > In summary, we have conducted additional experiments on video object segmentation tasks and demonstrated superior performance compared to existing state-of-the-art fine-tuning methods designed for Mamba architectures, **thereby providing stronger support for our claim of improved spatial modeling capabilities**. While we have not performed experiments on VIS and VPS tasks due to the limited availability of Mamba-based backbone models in these domains, we will continue to track relevant research developments and explore the application of our method to these tasks in future investigations.
> > >
> > > Table 1. Comparison on the VTUS dataset.
> > >
> > > | Method | Backbone | Param | Dice | Jaccard | Precision | Recall |
> > > |--------|--------|-------|------|---------|-----------|--------|
> > > | VPT | Vivim | 0.81M | 0.6670 | 0.5444 | *0.6721* | 0.8247 |
> > > | VFPT | Vivim | 0.81M | 0.6657 | 0.5435 | 0.6681 | 0.8279 |
> > > | SVP | Vivim | 1.54M | *0.6708* | *0.5495* | 0.6709 | *0.8365* |
> > > | **SSP** *(Ours)* | Vivim | 0.97M | **0.6873(+0.0165)** | **0.5696(+0.0201)** | **0.6843(+0.0122)** | **0.8420(+0.0055)** |
> > >
> > >
> > >
> > > [1] Yang, Yijun, et al. "Vivim: A video vision mamba for medical video segmentation." arXiv preprint arXiv:2401.14168 (2024).

---

> > > > ### Comment · Reviewer_VAWP · 2025-08-03
> > > >
> > > > The authors' rebuttal has addressed most of my concerns. I am currently considering raising my score and will make a final decision during the discussion phase.

---

> > > > > ### Author Response · Authors · 2025-08-03
> > > > >
> > > > > Dear reviewer VAWP:
> > > > >
> > > > > **Thank you for your thoughtful feedback and for considering raising your score based on our rebuttal.** We are grateful that our responses have addressed most of your concerns and appreciate your recognition of the improvements we have made. Your constructive comments were instrumental in helping us refine the presentation and strengthen the manuscript. We truly value the opportunity to clarify our approach and are thankful for the time and effort you invested in reviewing our submission.
> > > > >
> > > > > Best regards,
> > > > >
> > > > > Authors

---

### Official Review · Reviewer_Br1d · 2025-07-01

**Clarity:** 3
**Significance:** 3
**Originality:** 3
**Rating:** 4
**Confidence:** 3

**Summary:**

SSP is a PEFT method for Mamba-based video understanding models. It tackles the information decay in state space models (SSMs) through a "gathering and spreading" framework. This framework comprises Intra-Frame Gathering (IFG), which extracts salient spatial features and quantifies their importance via information entropy and spatial variance, and Inter-Frame Spreading (IFS), which uses this data to create global temporal prompts and gates temporal information based on spatial variance. SSP achieves competitive performance with minimal trainable parameters.

**Questions:**

In Table 2, the number of trainable parameters for SSP increases significantly with model size (e.g., from 0.98M in VideoMamba-S to 2.41M in -M), while STOP remains relatively stable despite also using attention and convolution layers. It would be helpful if the authors could provide a breakdown of where these additional parameters come from, and explain why the proposed method scales more heavily with model size compared to STOP.

**Ethical Concerns:**

["NO or VERY MINOR ethics concerns only"]

**Final Justification:**

The rebuttal clarifies most of the points I raised. Although there are still some limitations, as noted by other reviewers, I acknowledge the strengths of the work and will retain my initial evaluation.

**Limitations:**

yes

**Quality:**

3

**Strengths And Weaknesses:**

(+) The visualization of information decay in state space models is insightful and helps motivate the need for the proposed SSP framework.
(+) The proposed method demonstrates strong performance across multiple video understanding benchmarks, showing its practical effectiveness.

(-) While the paper states that the proposed modules are applied after the first Mamba layer, it remains unclear why this particular depth was chosen, and how the performance or parameter efficiency would change if the modules were applied at deeper layers. This is especially relevant for the IFG module, which computes entropy and spatial variance from intermediate representations — quantities that are likely to be sensitive to the semantic level of features. An ablation or discussion on the layer-wise placement of IFG/IFS would help clarify the design rationale and the robustness of the proposed method.
(-) The comparison with STOP could be misleading, as the two methods are evaluated under different objectives—STOP adopts a text-video matching paradigm, whereas SSP is evaluated using a classification loss. To more fairly assess the effectiveness of the proposed method, it would be helpful to report results under the same matching-based setup as STOP.

Minor
Figure 2 shows the proposed modules being inserted after the i-th layer, while the main text states that they are applied after the first Mamba layer. For clarity and consistency, it would be better if the figure also reflected the first-layer placement.

---

> ### Author Rebuttal · Authors · 2025-07-30
>
> ### W1 Response: The Placement Depth of IFG/IFS
>
> Thank you for your valuable suggestion. We conducted ablation experiments on the placement depth of IFG/IFS on the HMDB51 and Breakfast datasets, with results shown in the Table 1, which implies the following conclusions:
>
> (1) **Applying the modules to deeper layers does not yield performance improvements**. When our module is applied after the 8th layer, our method experiences slight accuracy decreases of **0.19%** and **0.01%** on HMDB51 and Breakfast datasets, respectively. When applied to deeper layers, the accuracy gradually declines. When our module is applied after the 24th layer, the accuracy significantly drops by **5.21%** on the Breakfast dataset and by **3.00%** on the HMDB51 dataset. *It is because that after sequential compression by multiple Mamba layers, some discriminative information in the token sequence becomes more difficult for IFG/IFS to extract.* While our method achieves optimal performance through appropriate hyperparameter selection, obtaining the highest accuracy of **76.66%** and **93.23%** on the HMDB51 and Breakfast datasets, respectively
>
> (2) **Applying the modules to deeper layers **does not** change the parameter efficiency**, as our IFG/IFS weights are shared across layers. As shown in Table 1, when SSP modules are applied at different depths, the trainable parameters are fixed at **2.41M**. Our method achieves effective aggregation and propagation of discriminative spatiotemporal information using only a small number of trainable parameters, surpassing existing state-of-the-art fine-tuning methods by an average of **2.76%** as demonstrated in Table 2 in the main text of our paper, *with the number of trainable parameters remaining constant regardless of placement depth, which further verifies our robustness.*
>
> Table 1. Ablation Experiments on the Placement Depth of IFG/IFS
>
> | Methods \ Layers | 1     | 8     | 16    | 24    |
> |------------------|-------|-------|-------|-------|
> | HMDB51          | **76.66** | 76.47(-0.19) | 75.55(-1.11) | 73.66(-3.00) |
> | Breakfast       | **93.23** | 93.22(-0.01) | 92.70(-0.53)  | 88.02(-5.21) |
>
> ### W2 Response: Comparison with STOP
>
> Thanks for your valuable suggestion. **We have reported results under the same matching-based setup with STOP.**
>
> In the original STOP paper, the authors performed evaluations in two paradigms: classification and matching.
>
> (1) As for the classification task, our method achieves accuracy improvements of **4.66%**, **1.73%**, and **32.32%** over STOP on the HMDB51, UCF101, and SS-V2 datasets respectively, as shown in Table 1 of the original STOP paper. Our method efficiently aggregates and propagates key information through fine-tuning designs tailored for the Mamba architecture, demonstrating superior performance on classification tasks, which further verifies the effectiveness of our SSP method in parameter-efficient fine-tuning on the Mamba architecture.
>
> (2) As for matching task, we have conducted more experiments on text-video matching tasks using MSVD dataset. As shown in Table 2, our method achieves an average recall rate that exceeds STOP by **9.06** in text-to-video retrieval and by **12.33** in video-to-text retrieval. As a prompt tuning method that similarly focuses on the Visual Encoder, **SSP demonstrates stronger text-video matching capability through efficient aggregation and propagation of discriminative spatiotemporal features**, which further showcases the effectiveness of SSP in parameter-efficient fine-tuning on the Mamba architecture.
>
> Table 2. Comparison results with STOP under the text-video matching paradigm
>
> | Methods | Visual Backbone | Pretrain Pairs | Params | Text→Video R@1 | Text→Video R@5 | Text→Video R@10 | Text→Video R@Avg | Video→Text R@1 | Video→Text R@5 | Video→Text R@10 | Video→Text R@Avg |
> |---------|-----------------|----------------|--------|----------------|----------------|------------------|------------------|----------------|----------------|------------------|------------------|
> | STOP    | VideoMamba      | 25M            | 1.5M   | 36.11          | 65.19          | 75.86            | 59.05            | 52.84          | 71.04          | 76.87            | 66.92            |
> | **SSP** (*Ours*)    | VideoMamba      | 25M            | 2.4M   | **45.62(+9.51)**          | **75.06(+9.87)**          | **83.66(+7.80)**            | **68.11(+9.06)**            | **66.12(+13.28)**          | **83.73(+12.69)**          | **87.91(+11.04)**            | **79.25(+12.33)**            |
>
> ### W3 Response: About Figure 2
>
> Thank you for your comments. As you said, our proposed modules are applied after the first layer and every subsequent Mamba layer. Therefore, we have supplemented Figure 2 with the insertion of our modules after the 1st layer to further clarify our method, which will be reflected in the camera-ready version. Thank you again for your valuable suggestions.
>
> ### Q1 Response: Where the additional parameters come from?
>
> **It should be clarified that the variation in the number of trainable parameters in our SSP stems from changes in the number of stacked IFS modules, rather than changes in model size.**
>
> *Firstly*, when applied to the smaller VideoMamba-S model, we only need to use **1** IFS module to surpass other fine-tuning methods, with a parameter count of **0.98M**. Stacking more modules does not yield performance improvements.
>
> *Secondly*, when applied to the larger VideoMamba-M model, as shown in the ablation experiment in Figure 7, the best performance is achieved with **3** stacked IFS modules, corresponding to a parameter count of **2.41M**. Following the original STOP paper, we did not stack the attention layers or convolutional layers of the STOP method, thus the parameters remained stable at **1.49M**.
>
> *In summary*, regardless of whether under the setting of fewer trainable parameters (**0.98M**) or more trainable parameters (**2.41M**), our method achieves average improvements of **1.21%** and **2.76%** compared to existing state-of-the-art fine-tuning methods in Table 1 and Table 2 in the main text, respectively, through the extraction of discriminative features from the hidden states of the Mamba model. **Under the same parameter efficiency, this demonstrates the effectiveness of our approach.**

---

> > ### Author Response · Authors · 2025-08-05
> >
> > Dear Reviewer Br1d,
> >
> > Thank you for your time and effort in reviewing our work. As the rebuttal period concludes within three days, we wish to respectfully confirm whether our responses have adequately addressed your initial concerns. Any feedback would be greatly appreciated.
> >
> > Sincerely,
> >
> > The Authors

---

### Official Review · Reviewer_TNYB · 2025-07-03

**Clarity:** 3
**Significance:** 3
**Originality:** 2
**Rating:** 5
**Confidence:** 4

**Summary:**

In this paper, a state space prompting (SSP), which is the prompt learning algorithm based on VideoMamba, proposed for video understanding. The proposed SSP consists of two core modules: an intra-frame gathering (IFG) module and an inter-frame spreading (IFS) module. IFG module generates the intra-frame prompts, which gathers the models’ attention to local features,  information entropy weights, and spatial variance measurements. IFS module generates the inter-frame prompts representing temporal inductive biases which spread the gathered temporal information across the frames. SSP only optimizes a few additional parameters and classification head during training. The extensive experimental results on various video understanding benchmarks demonstrate that the proposed SSP achieves better performances than conventional methods with small amounts of trainable parameters.

**Questions:**

I have carefully read the paper, but as I mentioned in the weaknesses section, I don’t think the proposed algorithm has any significant shortcomings. I believe this is a good paper that can make a meaningful contribution to the community. Nevertheless, I would like to offer a few suggestions below.

---

* In Tables 1 and 2, STOP shows significantly lower performances compared to other methods. It would be helpful to include additional explanation for these results. Although STOP is not specifically designed for VideoMamba, it is similar to SSP in that it is based on intra-frame and inter-frame prompts. Providing more insight into the reasons behind the significant performance gap would strengthen the discussion.

* This paper provides experimental results on four popular datasets, including the large-scale SSv2. Nevertheless, including performance comparisons on additional benchmarks such as Kinetics-400 or LVU would further strengthen the effectiveness of the proposed SSP.

**Ethical Concerns:**

["NO or VERY MINOR ethics concerns only"]

**Final Justification:**

During the rebuttal, the authors addressed my concerns well, and I appreciate all their efforts. I have not identified any additional weaknesses in the proposed algorithm, so I have decided to maintain my original rating of “accept.”

**Limitations:**

yes

**Quality:**

3

**Strengths And Weaknesses:**

> **Strength**
- This paper is well-written and easy to follow.
- The proposed SSP appears to be technically sound. It also provides a mathematical analysis of the proposed algorithm’s long-range information transmission capability.
- The proposed algorithm outperforms conventional algorithms on various benchmark tests.

> **Weakness**
- I don’t think there are any significant weaknesses in the proposed algorithm.

---

> ### Author Rebuttal · Authors · 2025-07-30
>
> ### Q1 Response: STOP shows significantly lower performances compared to other methods
>
> Thank you for your valuable suggestion. As discussed in Section 2.3 of this paper, the significant performance gap relies on two reasons:
>
> (1) Vision prompting models with Transformer architecture leverage global attention mechanisms that allow tokens at arbitrary positions in the sequence to interact, whereas Mamba-based models process tokens sequentially, leading to adjacent tokens tending to contain more overlapping information. **STOP inputs all tokens into the inter-frame prompting module for computation, introducing excessive redundant information when modeling contextual relationships**, a phenomenon that is more pronounced in video data. In contrast, other fine-tuning methods do not simultaneously model all tokens, making them relatively more capable of extracting key discriminative information from tokens that have been highly compressed by Mamba compared to STOP.
>
> (2) As shown in Table 1, when based on Transformer models, this method exhibits similarly low performance on the large-scale dataset SSv2 as when based on Mamba models, **indicating that in the efficient fine-tuning domain with limited trainable parameter space, greater computational overhead does not necessarily lead to better performance**. Reasonable inductive biases in method design, such as the complementary information aggregation and propagation mechanisms in SSP, can yield superior results.
>
> Compared with STOP, our method utilizes the IFS module to generate inter-frame prompts through token sampling while aggregating local spatial features via IFG, and precisely regulates the influence of local features on global temporal features through entropy weight and spatial variance, thereby learning discriminative spatiotemporal information during fine-tuning. Therefore, as shown in Table 1, our method achieves a **30.28%** improvement on the large-scale dataset SSV2, which verifies that our approach more effectively adapts to the Mamba architecture during fine-tuning.
>
> Table 1. Comparison with STOP
>
> | Method | Backbone | HMDB51 | SSV2 | UCF101 | Avg |
> |--------|----------|--------|------|--------|-----|
> | STOP | VisionTransformer-B | *72.00* | 21.40 | *95.30* | 62.90 |
> | STOP | VideoMamba-M | 71.96 | *23.44* | 94.60 | *63.33* |
> | **SSP** *(Ours)* | VideoMamba-M | **76.66(+2.66)** | **53.72(+30.28)** | **97.03(+1.73)** | **75.80(+12.47)** |
>
> ### Q2 Response: Comparisons on more benchmarks
>
> Thanks for your valuable suggestion. **We have conducted additional experiments as follows:**
>
> (1) For the Kinetics-400 benchmark, since our VideoMamba is pre-trained on Kinetics-400, where high zero-shot performance of **83.3%** has already been achieved, and existing fine-tuning work typically does not test on pre-training datasets [1][2], we did not conduct further fine-tuning experiments on Kinetics-400 to avoid redundancy.
>
> (2) For the LVU benchmark, we conduct experiments on the LVU benchmark, as shown in Table 2 below, SSP achieves an average performance improvement of **3.64%** over existing state-of-the-art methods on LVU through efficient aggregation and propagation of discriminative spatiotemporal information, which further confirms the effectiveness of our approach.
>
>
> Table 2. Comparisons on LVU
>
> | Methods | Backbone | Params | Rel. | Speak. | Genre | Wtr | Avg |
> |---------|----------|--------|------|--------|-------|-----|-----|
> | Adapter | VideoMamba-Ti | 0.59M | 53.65 | *32.44* | *52.32* | 29.76 | 42.04 |
> | VFPT | VideoMamba-Ti | 0.27M | 56.09 | 31.38 | 51.97 | 26.19 | 41.40 |
> | SVP | VideoMamba-Ti | 0.63M | *56.81* | *32.44* | 52.22 | *33.33* | *43.70* |
> | **SSP** *(Ours)* | VideoMamba-Ti | 0.21M | **58.53(+1.72)** | **34.89(+2.45)** | **53.70(+1.38)** | **42.26(+8.93)** | **47.34(+3.64)** |
>
> [1]: Pan, Junting, et al. "St-adapter: Parameter-efficient image-to-video transfer learning." Advances in Neural Information Processing Systems 35 (2022): 26462-26477.
>
> [2]: Liu, Zichen, et al. "STOP: Integrated Spatial-Temporal Dynamic Prompting for Video Understanding." Proceedings of the Computer Vision and Pattern Recognition Conference. 2025.

---

> > ### Author Response · Authors · 2025-08-05
> >
> > Dear Reviewer TNYB,
> >
> > Thank you for your time and effort in reviewing our work. As the rebuttal period concludes within three days, we wish to respectfully confirm whether our responses have adequately addressed your initial concerns. Any feedback would be greatly appreciated.
> >
> > Sincerely,
> >
> > The Authors

---

> > > ### Comment · Reviewer_TNYB · 2025-08-06
> > >
> > > Thank you for the response. I also carefully read other reviews including negative ones and response. In overall, I think the responses from authors have well addressed my concerns and also the concerns raised by the other reviewers. Since I've found no additional concern, I will keep my original rating (Accept).

---

> > > > ### Author Response · Authors · 2025-08-06
> > > >
> > > > Dear Reviewer TNYB,
> > > >
> > > > Thank you for your response and for your thoughtful consideration of our work throughout this review process. We sincerely appreciate you taking the time to carefully read through the other reviews and our rebuttal. Your positive assessment and decision to maintain your original rating is a great encouragement to us.
> > > >
> > > > We are very grateful for your support and constructive feedback throughout this process.
> > > >
> > > > Best regards,
> > > >
> > > > The Authors

---

### Note · Authors · 2025-08-12

Dear Reviewers, Area Chairs, and Program Chairs of NeurlPS,

We would like to express our most sincere gratitude to all the reviewers, Area Chairs, and Program Chairs. Your valuable suggestions have been crucial in the further improvement of our paper.

Firstly, we are very grateful to Reviewers **Br1d** and **Ab59** for their recognition of our research motivation. Meanwhile, we sincerely appreciate reviewers **TNYB** and **VAWP** for their recognition of the technical design of our algorithm. To achieve efficient fine-tuning of state-space models for video tasks, we introduced the IFG and IFS modules to effectively aggregate and propagate discriminative spatio-temporal information. Your valuable acknowledgment has significantly enhanced the novelty of our work.

Secondly, in response to the concerns raised by Reviewer **Ab59** regarding the novelty and operational mechanism of our method, we have provided a thorough discussion in our rebuttal. In addition, to address the questions from Reviewer **VAWP** concerning the modeling of fine-grained spatio-temporal information, we conducted additional experiments on temporal action segmentation, temporal action localization, and video object segmentation, providing further validation and analysis. Combining the above clarifications, both Reviewer **Ab59** and Reviewer **VAWP** have acknowledged the novelty, operational mechanism, and fine-grained spatio-temporal modeling capabilities of our method. Consequently, both **Ab59** and **VAWP** have indicated that they will raise their scores for our paper. Furthermore, Reviewer **Ab59** had no further questions following our comprehensive response. We are extremely grateful for the efforts of all reviewers, which have substantially improved the quality of our paper.

In summary, the issues raised by the reviewers do not fundamentally challenge our core ideas, methodological novelty, or experimental analysis, nor do they necessitate substantial revisions to the manuscript. Nevertheless, we commit to addressing all reviewers' valuable comments comprehensively in the final manuscript.

Finally, we wish to once again express our deepest appreciation to all the reviewers, Area Chairs, and Program Chairs.

Sincerely,

The Authors of Paper 3493

---

### Decision · Program_Chairs · 2025-09-17

**Decision:**

Accept (poster)

**Comment:**

All reviewers commend the intuitive motivation behind the proposed state-space prompting method for enhancing inter- and intra-frame interactions in video understanding tasks. The visualization of information decay in state space models provides insightful justification for the proposed SSP framework and effectively motivates its design. The authors' rebuttal successfully addressed initial concerns regarding the limitation to video classification tasks, unfair comparisons to STOP, missing ablations on the placement of IFG/IFS modules, and questions about the work's novelty. All reviewers expressed satisfaction with the responses to their points and indicated willingness to increase their ratings, with several following through on this commitment. The area chairs concur with the reviewers' assessment that this submission makes a valuable contribution to the field of video understanding, presenting a well-motivated approach with solid experimental validation and thorough analysis of the proposed components.